# Graph-Supported Dynamic Algorithm Configuration for Multi-Objective Combinatorial Optimization

**Robbert Reijnen** [1]   **Yaoxin Wu** [1]   **Zaharah Bukhsh** [1]   **Yingqian Zhang** [1]

## Abstract

Deep reinforcement learning (DRL) has been widely used for dynamic algorithm configuration, particularly in evolutionary computation, which benefits from the adaptive update of parameters during the algorithmic execution. However, applying DRL to algorithm configuration for multi-objective combinatorial optimization (MOCO) problems remains relatively unexplored. This paper presents a novel graph neural network (GNN) based DRL to configure multi-objective evolutionary algorithms. We model the dynamic algorithm configuration as a Markov decision process, representing the convergence of solutions in the objective space by a graph, with their embeddings learned by a GNN to enhance the state representation. Experiments on diverse MOCO challenges indicate that our method outperforms traditional and DRL-based algorithm configuration methods in terms of efficacy and adaptability. It also exhibits advantageous generalizability across objective types and problem sizes, and applicability to different evolutionary computation methods.

## 1. Introduction

Selecting the right hyperparameters is crucial for the performance of optimization algorithms. Some automated algorithm configuration (AC) methods (López-Ibáñez et al., 2016; Lindauer et al., 2022) have been developed to identify well-performing configurations and reduce the need for labor-intensive trial-and-error tuning. As the optimal parameter values may change throughout different stages of algorithmic deployment (Aleti, 2012), various dynamic algorithm configuration (DAC) methods have been proposed in recent years (Biedenkapp et al., 2020; Adriaensen et al.,

2022). DAC adjusts the configuration of algorithms in real time, which is advantageous for algorithms facing changes in the search space configuration during execution. This adaptability is especially relevant for iterative algorithms, such as Evolutionary Algorithms (EAs), a prominent class of Evolutionary Computation (EC) techniques for solving complex optimization problems. The performance of EAs relies on the precise adjustment of their parameters and may require changes at various phases of the search process to maintain optimal performance.

Deep Reinforcement Learning (DRL) has been successfully used to control parameter values for various single-objective EC algorithms, as reported in the literature (Sharma et al., 2019; Reijnen et al., 2024; Begnardi et al., 2025). These approaches address the parameter configuration problem by modeling it as a contextual Markov decision process (MDP) (Biedenkapp et al., 2020). This enables dynamic algorithm configuration to be approached as a sequential decision-making problem, enabling DRL to control algorithm configurations during search. Xue et al. (2022) extend the existing DRL-based DAC approaches to address multi-objective optimization. Although these methods have demonstrated their effectiveness in configuring parameters during the search, their applications are primarily limited to (multi-objective) continuous optimization, such as tuning hyperparameters of machine learning models, as in AutoML (Biedenkapp et al., 2020; Eimer et al., 2021), and benchmarking continuous functions (Xue et al., 2022).

In this paper, we propose a DRL-based, dynamic algorithm configuration method designed specifically for solving multi-objective combinatorial optimization problems (MOCOs). Most (multi-objective) combinatorial optimization problems, such as machine scheduling, vehicle routing, and resource allocation problems, are NP-hard, as they involve finding high-quality solutions in a large space of discrete decision variables. Hence, practical approaches for solving these problems typically rely on heuristics, among which EAs have been widely used in various COPs (Bartz-Beielstein et al., 2014; Zhou et al., 2011).

We expect (and confirm with the experiments in Section 4.1) that the existing DAC approach designed for continuous optimization (i.e., MADAC (Xue et al., 2022)) may not work

---

[1]Department of Industrial Engineering and Innovation Sciences, Eindhoven University of Technology, Eindhoven, The Netherlands. Correspondence to: Yaoxin Wu <y.wu2@tue.nl>.

*Proceedings of the 42nd International Conference on Machine Learning*, Vancouver, Canada. PMLR 267, 2025. Copyright 2025 by the author(s).

well on large-sized, complex COPs with many objectives, due to less smooth solution spaces and a wide range of objective values of COPs. To address these challenges, our proposed method, called GS-MODAC, employs a Graph Neural Network to capture the state of the search algorithm. Specifically, we take inspiration from various convergence- and diversity–based metrics for multi-objective optimization, such as the number of elite solutions, the spacing between solutions, the relative size of holes (gaps) in the solution space, and hypervolume. With this, we expect that our method leverages the graph-based representation to dynamically learn similar (yet advanced) features during the optimization process to reflect the current state in the multiple objective planes. By representing the state space as a graph, our method provides a state configuration independent of the number of objectives, eliminating the need for practitioners to configure arbitrary state features manually. In addition, GS-MODAC leverages a rewarding scheme designed to be incentivized toward Pareto optimal solutions in a problem-agnostic manner, fostering generalizability between differently scaled COPs.

Experimentation demonstrates that GS-MODAC is better than state-of-art algorithm configuration methods based on heuristics (irace) (López-Ibáñez et al., 2016), Bayesian Optimization (SMAC3) (Lindauer et al., 2022), and a multi-agent DRL approach (MADAC) (Xue et al., 2022). We further demonstrate that the proposed method can be applied to multiple Multi-Objective Evolutionary Computation algorithms to solve different MOCOs from distinct problem domains featuring varying numbers of objectives. Also, the trained models can generalize to effectively solve instances of larger sizes and more constrained problem variants, which were not observed in training.

Our study offers the following contributions:

1) We introduce GS-MODAC, a GNN and DRL-based method for dynamically controlling MOEA parameter configurations for solving MOCOs. This approach overcomes the limitations of static algorithm configuration methods, achieving better convergence and more diverse solutions.

2) We propose a graph representation of solutions in the objective space, which is learned by graph neural network and involved in the state. Based on the normalized objectives, we also present an instance-agnostic reward function that applies to problems of different types and varying sizes.

3) We evaluate the proposed method on routing and scheduling problems and demonstrate its promising generalizability to perform effectively on more constrained problem variants and larger problem instances unseen during training.

## 2. Background and Related Work

**Multi-Objective Optimization (MOO) and Combinatorial Optimization (MOCO).** Combinatorial Optimization is concerned with finding the best solution from a finite set of feasible solutions. These problems are characterized by their discrete nature, where the solutions can be represented as integers, graphs, sets, or sequences. Multi-Objective Combinatorial Optimization (MOCO) involves simultaneously optimizing multiple, often conflicting objectives for combinatorial optimization problems. The general formulation of MOCO can be expressed as $\min_{x \in X} f(x) = (f_1(x), f_2(x), \ldots, f_N(x))$. Here, $X$ denotes the set of feasible solutions, $N$ is the number of objective functions to be optimized, and each $f_i(x)$ represents an objective function to be minimized.

**Definition 1: Pareto Dominance.** A solution $x_1 \in X$ dominates another solution $x_2 \in X$ ($x_1 \prec x_2$) if and only if: $f_i(x_1) \leq f_i(x_2)$ for all $i \in \{1, \ldots, N\}$, and there exists at least one $j \in \{1, \ldots, N\}$ such that $f_j(x_1) < f_j(x_2)$.

**Definition 2: Pareto Optimality.** A solution $x^* \in X$ is considered Pareto optimal if there is no other solution $x' \in X$ satisfying $x' \prec x^*$. In other words, $x^*$ is Pareto optimal if it is not dominated by any other solution in $X$.

**Definition 3: Pareto Front.** The objective of multi-objective optimization is to find the Pareto front, which consists of all Pareto-optimal solutions: $\mathcal{P} = \{x^* \in X \mid \nexists x' \in X \text{ such that } x^* \prec x'\}$. The corresponding Pareto front is defined as: $\mathcal{F} = \{f(x) \mid x \in \mathcal{P}\}$. The Pareto front consists of the objective values of the Pareto set, where each $f(x)$ represents a point in the objective space.

**Definition 4: Hypervolume Indicator.** The Hypervolume (HV) indicator is a widely used metric for assessing performance in multi-objective optimization problems, providing a comprehensive evaluation of both convergence and diversity, even without knowledge of the exact Pareto front (Zitzler & Thiele, 1998). For a Pareto front $\mathcal{F}$ in the objective space, the HV with respect to a fixed reference point $r \in \mathcal{R}^N$ is defined as:

$$\text{HV}_r(\mathcal{F}) = \mu \left( \bigcup_{f(x) \in \mathcal{F}} [f(x), r] \right) \quad (1)$$

where $\mu$ is the Lebesgue measure, representing the $N$-dimensional volume, and $[f(x), r]$ refers to an $N$-dimensional cube: $[f(x), r] = [f_1(x), r_1] \times [f_N(x), r_N]$, spanning the region in the objective space between a point on the Pareto front and a fixed reference point $r$.

**Algorithm Configuration.** Algorithm Configuration (AC) involves determining optimal parameter configurations for an algorithm to maximize performance across various inputs. Dynamic Algorithm Configuration (DAC) extends AC by adjusting parameters during the optimization process to enhance performance (Biedenkapp et al., 2020). Unlike static configurations, DAC aims to balance exploration and exploitation, increasing the likelihood of finding high-quality solutions. According to Karafotias et al. (2014), it can be classified into three types: 1) *Deterministic*, which changes parameter configurations based on a predetermined rule, often using a time-varying schedule (Sun et al., 2020); 2) *Self-adaptive*, integrating parameter adjustments into the search process, allowing parameters to evolve alongside solutions (Michalewicz et al., 2000); and 3) *Adaptive parameter control*, which adjusts parameters based on search feedback, using credit assignment and operator selection to optimize performance (Aleti & Moser, 2016).

Machine Learning methods like Bayesian Optimization and Neural Networks have been used to tune parameters by predicting parameter performances based on training instances (Lessmann et al. (2011); Biswas et al. (2021); Centeno-Telleria et al. (2021)). Recently, Reinforcement Learning (RL) has gained attention for dynamic algorithm configuration, especially in evolutionary algorithms (EAs), where parameter configurations can act as actions, and when a configuration set leads to improved solutions, a reward is given to the RL agent. Recent research has demonstrated the effectiveness of RL in controlling the parameters of EAs. For example, Q-learning has been applied to adapt the crossover and mutation rates of each generation to solve a vehicle routing problem (Quevedo et al., 2021). Similarly, an EA has been hybridized with SARSA and Q-Learning to control crossover and mutation rates for the flexible job-shop scheduling problem (Chen et al., 2020). Building on RL, Deep Reinforcement Learning (DRL) has demonstrated considerable potential. Notable examples include the use of a double deep Q-Network agent (DDQN) to select parameters in Differential Evolution (DE) (Sharma et al., 2019) or a Policy Gradient method (Sun et al., 2021), and the works of (Shala et al., 2020; Speck et al., 2021; Biedenkapp et al., 2022) which have shown to generalize to longer horizons and fairly well to larger problem dimension. Extensions of these approaches have further explored multi-objective optimization (Huang et al., 2020; Tian et al., 2022; Reijnen et al., 2022; 2023b). Despite these advancements, applying DRL in multi-objective optimization presents several challenges. Many existing approaches rely on manually configured features derived from convergence and fitness landscapes, such as the number of elite solutions, solution spacing, the relative size of gaps in the solution space, and hypervolume, to define the states in the MDP. This process is labor-intensive and often suboptimal. Additionally, manag-

ing high-dimensional configured state spaces and optimizing for multi-objectives complicates the learning process (Yang et al.). Moreover, most studies focus on search operator selection, typically configured as discretized actions, and are often trained and demonstrated on simple continuous optimization problems and standard benchmark functions (Ma et al., 2024).

The closest work to ours is Xue et al. (2022), where the authors propose MADAC for tuning parameters in a multi-objective evolutionary algorithm (MOEA). The work utilizes value-decomposition networks (VDN) (Sunehag et al., 2017), a typical multi-agent RL method, to identify the optimal settings for different categories of parameters. The work incorporates information from the specific problem instance, the ongoing optimization process, and the evolving population of solutions. The reward function incentivizes improvement, offering rewards for discovering better solutions and greater rewards for further advancements in later stages. The limitation of MADAC is that it typically includes information on convergences, objectives, and population-based metrics based on arbitrary hand-defined and tuned state features. This reliance on manually selected features can lead to suboptimal results, as the chosen features may not adequately capture the complexity of the environment. To address this, we propose a novel DRL-based approach for the dynamic configuration of parameters in MOEAs aimed at solving Multi-objective Combinatorial Optimization problems. Instead of relying on arbitrarily defined features, our approach involves mapping the objective spaces to graph structures and utilizing Graph Neural Networks (GNNs) to aggregate node features as states. This method allows for a more comprehensive and adaptive representation of the state space, scalable to multiple objective problems, potentially enhancing the performance and robustness of the EAs.

## 3. The Method

This section presents our proposed method, GS-MODAC (Graph-Supported Multi-Objective Dynamic Algorithm Configuration). GS-MODAC employs a Graph Neural Network (GNN) to capture the state of the search algorithm and Deep Reinforcement Learning (DRL) to configure the next search iteration to solve MOCOs. Graphs provide a flexible way to represent structured embeddings, and GNNs effectively model complex structures and extract meaningful representations (Zhou et al., 2020). In this work, GNNs extract the graph state, allowing the DRL agent to make more informed decisions based on the current search state. By representing the state space as a graph, our method uses a state configuration independent of the number of objectives, eliminating the need for practitioners to manually customize state representations for varying numbers of objectives. We illustrate the overview of GS-MODAC in Figure 1.

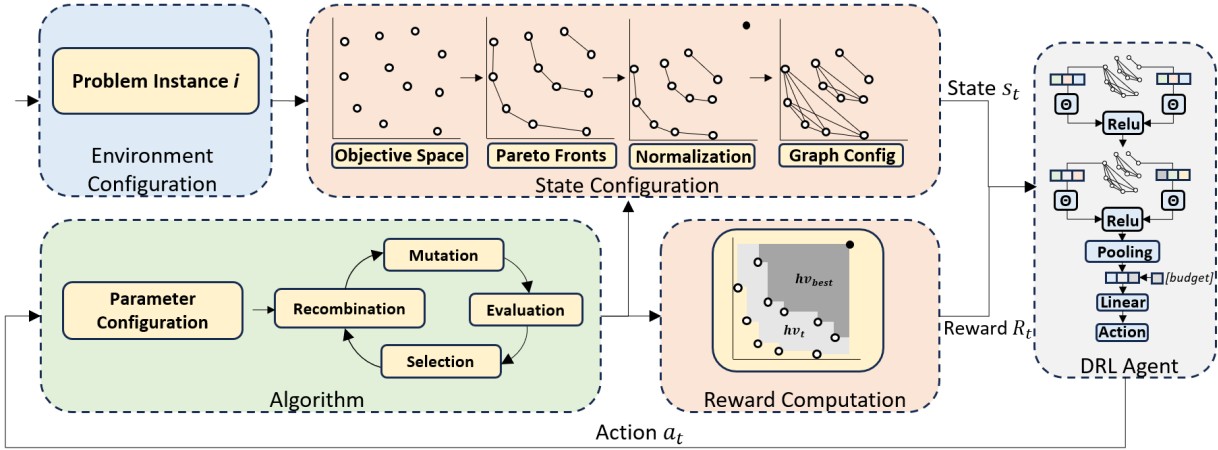

Figure 1: The GS-MODAC framework. The framework integrates a DRL agent with a graph-based representation of the search state to dynamically configure each iteration of the search for multi-objective optimization. At each iteration, the current population is converted into a graph where nodes represent normalized objective values of solutions across multiple objective planes, and edges connect solutions within the different Pareto fronts. A Graph Neural Network (GNNs) extracts an embedding, which the agents uses configure the next iteration. The chosen actions are applied to the environment, which returns a new population of solutions, an updated state, and a reward signal for learning.

### 3.1. MDP formulation for GS-MODAC

GS-MODAC is built upon the foundation of Dynamic Algorithm Configuration (DAC) principles (Biedenkapp et al., 2020), dynamically adjusting parameter configuration of EAs during their optimization processes. This process can be formulated as a contextual Markov Decision Process (MDP) $M_I$, with shared action and state spaces, but with different transition and reward functions for each instance $i$ in $I$. Each $M_i$ corresponds to the MDP of a specific problem instance $i$, encapsulating the state space $S$, action space $A$, state transition function $T_i$, and reward function $R_i$.

In GS-MODAC, given a target algorithm with a configuration space $\Theta$, a policy $\pi$ maps a state $s \in S$ to an action $a \in A$, corresponding to a hyperparameter configuration $\theta \in \Theta$. The goal is to train a policy that enhances algorithm performance across a diverse set of problem instances by minimizing the expected cost $c(\pi, i)$ over instances $i \in I$. To further facilitate generalizability, we define a shared reward function $R$ to consistently measure performance improvement across different problem instances. This function $R$ ensures that the policy learns to optimize the performance of the target algorithm to generalize well across various instances rather than overfitting to specific instances. We introduce the components of the MDP underlying GS-MODAC as follows:

**States.** The state space $S$ provides a DRL agent with information on the current status of the search algorithm, helping to select the best action for the next iteration. Several studies have attempted to create a state configuration that accurately represents the convergence process and general-izes to unexplored problem instances (Sharma et al., 2019; Sun et al., 2020; Xue et al., 2022). These configurations typically include convergence information, objective values, and population diversity metrics. In contrast to the literature, we innovatively propose mapping objective spaces to graphs and leveraging GNNs to dynamically learn state representations. The graph transformation of objective space is illustrated in 'state configuration' in Figure 1.

This transformation constructs a graph-based representation of the current population's solutions across multiple objective planes. Each node in the graph corresponds to a solution, with its features comprising only the normalized objective values. Normalization is performed using the best objective values encountered during the search and the worst values observed in the initial generation. Non-Dominated Sorting is applied to rank the nodes into distinct Pareto fronts. Nodes within the same front are then interconnected, resulting in a structured graph that captures the hierarchy of solutions. This eliminates the need for manual state space design, a process known to be cumbersome and suboptimal. To ensure the state configuration is independent of the magnitude of objective values, we normalize the solution space relative to a reference point that is defined by the worst observed objective values in the first population of solutions. In doing so, we provide a state configuration that effectively represents the algorithmic convergence and the diversity of solution performances, potentially generalizing to problem instances with varying objective magnitudes. An additional feature vector is correspondingly included, containing the normalized number of generations that have been passed by, representing the remaining budget available for the search.

**Actions.** The action space $A$ is represented by multiple continuous values, each associated with an evolutionary algorithm parameter to be controlled. These values are normalized between -1 and 1 on a parameter scale, defined based on the recommended values from rules of thumb for EA tuning (Coello et al., 2007).

**Transitions.** The transition function outlines the dynamics of the search algorithm and is led by interactions between the agent and the problem environment. In the context of GS-MODAC, each interaction (step) with the environment serves as a search iteration. Given state $s_t$, an agent takes a action $a_t$, and the probability of moving to state $s_{t+1}$ is denoted as $T(s_{t+1}|s_t, a_t)$. Unlike the state, action, and reward spaces (in the scope of this work), the transition function is contingent upon the specific instance $i \in I$.

**Rewards.** The reward function is critical to guide policy learning. In multi-objective optimization, the rewarding system should encourage algorithmic convergence towards the optimal Pareto front. However, the evolving towards the Pareto front often turns increasingly demanding along search steps. The early search stages typically allow for quick gains, while the later stages require substantially more effort. In light of this, we design rewards for enhancing the evolvement of the Pareto front in the latter.

In particular, we design the reward function as follows: At each iteration $t$, we assess whether the hypervolume of the population $HV_{current}$ exceeds the best hypervolume previously observed $HV_{best}$. If $HV_{current} > HV_{best}$, we compute the percentage improvements, that is, $\Delta_{current}$ and $\Delta_{best}$, and then calculate the reward as the difference between the squared improvements. In this way, we magnify the rewards for larger improvements in the later stages of the convergence, encouraging significant evolvement of the Pareto front. The reward is defined as follows:

$$
r_t = \begin{cases} \Delta_{current}^2 - \Delta_{best}^2 & \text{if } HV_{current} > HV_{best} \\ 0 & \text{otherwise} \end{cases}
$$

where $\Delta_{current}$ and $\Delta_{best}$ are calculated as follows:

$$
\Delta_{current} = \left( \frac{HV_{current} - HV_{initial}}{HV_{ideal} - HV_{initial}} \right) \times 100,
$$
$$
\Delta_{best} = \left( \frac{HV_{best} - HV_{initial}}{HV_{ideal} - HV_{initial}} \right) \times 100.
$$

Hypervolumes are calculated using a nadir point, defined by the worst-case values of objectives in the initial population of solutions. The ideal hypervolume $HV_{ideal}$ is computed using this nadir point, along with an ideal point, which is approximated by running the underlying evolutionary

algorithm one-time with a higher budget (e.g., doubled). It is worth noting that our reward function is instance-agnostic and thus applicable to different instances of varying sizes and complexities. We empirically observe that the reward function performs consistently well on different problems and delivers outstanding generalizability of trained models.

### 3.2. Graph-based policy learning and Training Algorithm

We use the Proximal Policy Optimization (PPO) algorithm (Schulman et al., 2017) to train the parameterized policy. PPO is a widely used and highly effective policy gradient algorithm that uses a probability ratio between policies to maximize the improvement of the current policy without the risk of performance collapse. In our case, the agent utilizes a neural network that first processes the representation of the graph-based state through two layers of the graph convolutional network (GCN) (Kipf & Welling, 2016). These layers are designed to extract and aggregate node embeddings (i.e., representations) effectively, capturing the essential structural information within the graph. Then, a global mean pooling operation is applied to average the node embeddings, producing a single embedding across the entire graph. The embedding is concatenated with an additional feature vector containing specific search budget information. The enriched embedding is finally fed into a linear layer to predict the mean values of action distributions. We have performed an ablation study in Appendix E, where we evaluated and tested the setup and verified the effectiveness of our approach.

### 3.3. Multi-Objective Evolutionary Algorithm Deployment

Exact methods can find the Pareto set in Multi-Objective Combinatorial Optimization (MOCO). However, the computational demands of these methods tend to increase exponentially with problem complexity, which often makes them impractical for large-scale applications. As a more feasible alternative, heuristic methods, particularly multi-objective evolutionary algorithms (MOEAs), are popular in practice due to their ability to effectively approximate Pareto fronts in a computationally efficient manner. In this work, we demonstrate GS-MODAC by applying it to two widely used algorithms: 1) NSGA-II (Deb et al., 2002), which implements a non-dominated sorting mechanism with a crowding distance metric to preserve solution diversity throughout the search, ensuring comprehensive exploration of the Pareto front; and 2) Multi-Objective Particle Swarm Optimization (MOPSO) (Coello & Lechuga, 2002), a swarm intelligence algorithm, which adjusts positions of particles by tracking both individual best locations and the best discoveries in the swarm. It integrates an archive to store non-dominated solutions to effectively cover the Pareto front.

# 4. Experiments

We evaluated GS-MODAC on two multi-objective combinatorial optimization problems to assess its performance, scalability, and generalization. Comparisons are made against static and dynamic algorithm configuration methods. All code and experimental details are made publicly available [1].

**Problems.** We apply our proposed method to two multi-objective combinatorial optimization problems: Flexible Job Shop Scheduling Problem (FJSP) and Capacitated Vehicle Routing Problem (CVRP). The FJSP involves scheduling multiple jobs, each composed of various operations, onto a set of machines. The operations of each job must be completed in a specific sequence, with each operation featuring a predefined processing time on specific machines. Based on the literature (Tamssaouet et al., 2022), we focus on minimizing Makespan, Balanced Workload, Average Flowtime, Total Workload, and Maximum Flowtime. We refer to the variants of FJSP as the Bi-, Tri- and Penta-FJPS, solving the first 2, first 3, and all 5 objectives, respectively. CVRP involves determining optimal routes for a fleet of vehicles to serve a set of customers. Each customer has a specific demand and each vehicle has a capacity limit that must not be exceeded. The objectives are to minimize the total travel distances and the longest route. We refer to the CVRP problem composed of these two objectives as the Bi-CVRP problem. Please refer to Appendix A for a comprehensive discussion of FJSP and CVRP, including the constraints and objectives addressed in this work.

**Instance generation.** For FJSP, we generate train and test instances for three distinct problem sizes: 1) 5 jobs and 5 machines (5j5m), 2) 10 jobs and 5 machines (10j5m), and 3) 25 jobs and 5 machines (25j5m), following the instance generation configuration of Song et al. (2022). We generate 200 instances for each size, consisting of 100 instances for training and 100 for testing. Each instance contains a varying number of operations per job, ranging from 4 to 8, and the processing time for each operation varies between 2 and 20 time units. The same instance sets are used for experiments with 2, 3, or 5 objectives. For CVRP, we generate 3 distinct sizes of 100, 200, and 500 customers, according to the instance generation of da Costa et al. (2021). We create 200 instances per problem size using random 2-dimensional coordinates for each customer and the depot in the 0 to 1 range. Each customer has a random demand between 1 and 9, and the vehicles have a capacity of 40 units.

**Baselines.** To show the performance of our proposed dynamic algorithm configuration method on solving multi-objective FJSP and CVPR, we use NSGA-II as a base algorithm, whose values have been configured with rules of thumb, configuring the crossover parameter as 0.7 and the mutation parameter as 0.02 (Coello et al., 2007). Additionally, as shown in Appendix D, we empirically validate that our method effectively configures MOPSO, a swarm intelligence-based approach. We compare the proposed GS-MODAC against three algorithm configuration methods for tuning NSGA-II parameters: two widely used static AC methods, SMAC3 (Lindauer et al., 2022) and irace (López-Ibáñez et al., 2016), and a recent RL-based DAC approach, MADAC (Xue et al., 2022).

SMAC3 is a hyperparameter tuning method that combines Bayesian optimization and random forest regression. For the tuning, we use the generated test instances for each given instance size. Bayesian optimization is used to draw parameter configurations from the defined parameter configuration ranges and evaluate them on the provided tuning instances over 10.000 runs of the NSGA-II configured algorithm, lasting between 5 to 14 hours for the Bi-CVRP instances and between 8 to 40 hours for the FJSP-variants. We also use the Iterated Race (irace) tuning method, which employs an iterative racing procedure. In each iteration (or 'race'), the worst-performing configurations are replaced with new ones, optimizing settings based on a set of given instances. irace was tuned with the same budget as the BO tuning method, taking between 3 and 12 hours for Bi-CVRP problem configurations and 5 to 20 hours for the FJSP-based variants, respectively. Since MADAC is designed to select discrete actions, we discretize the parameter space of NSGA-II with 10 actions between 0.6 and 1.0 as crossover rate and between 0 and 0.1 for the mutation rate (in line with rules-of-thumb for EA parameter configurations (Coello et al., 2007)).

**Training.** We trained GS-MODAC for each problem configuration with randomly generated problem-instance sizes. The action space for NSGA-II is defined as two continuous actions with ranges $\langle 0.6, 1.0 \rangle$ and $\langle 0.0, 0.1 \rangle$ for the crossover and mutation rates of NSGA-II. The training process involved 1.000,000 steps for the scheduling problems and 2.500.000 steps for the routing, configured with 50 generations of search and a population size of 50. It was carried out on an Intel(R) Core(TM) i7-6920HQ CPU @ 2.90GHz with 8.0GB of RAM and five parallel environments. The training duration varied for different-sized instance sets, taking around 11, 15, and 26 hours for the Bi-CVRP problem configurations and between 5 hours and 3 days for the different configured FJSP-based problems. The training process spans 2000 epochs with 500 steps per epoch. The model parameters are set as Schulman et al. (2017), and the network layers are configured with 64 nodes. The MADAC baseline model is trained according to Xue et al. (2022), taking between 2 and 8 hours for Bi-CVRP and 12 and 60 hours for the FJSP-based variants.

---

[1] https://github.com/RobbertReijnen/GS-MODAC

Table 1: Performance comparison of different methods in solving 100 instances of various problems of varying sizes 10 times, based on the mean found hypervolume (mean), the best-found hypervolume (max), and the standard deviation (std).

| | Bi-FJSP - 5j5m | | | Bi-FJSP - 10j5m | | | Bi-FJSP - 25j5m | | |
|---|---|---|---|---|---|---|---|---|---|
| Method | mean | max | std | mean | max | std | mean | max | std |
| NSGA-II | $1.87{\times}10^4$ | $2.02{\times}10^4$ | $1.21{\times}10^3$ | $3.82{\times}10^4$ | $4.11{\times}10^4$ | $2.29{\times}10^3$ | $9.41{\times}10^4$ | $9.93{\times}10^4$ | $4.84{\times}10^3$ |
| irace | $\mathbf{1.92{\times}10^4}$ | $\mathbf{2.04{\times}10^4}$ | $1.06{\times}10^3$ | $3.90{\times}10^4$ | $4.11{\times}10^4$ | $1.95{\times}10^3$ | $9.52{\times}10^4$ | $9.97{\times}10^4$ | $4.03{\times}10^3$ |
| SMAC3 | $1.91{\times}10^4$ | $2.04{\times}10^4$ | $1.09{\times}10^3$ | $3.89{\times}10^4$ | $4.13{\times}10^4$ | $2.19{\times}10^3$ | $9.51{\times}10^4$ | $9.97{\times}10^4$ | $4.46{\times}10^3$ |
| MADAC | $1.82{\times}10^4$ | $1.95{\times}10^4$ | $7.53{\times}10^2$ | $3.69{\times}10^4$ | $3.98{\times}10^4$ | $4.47{\times}10^3$ | $9.24{\times}10^4$ | $9.72{\times}10^4$ | $3.09{\times}10^3$ |
| GS-MODAC | $\mathbf{1.92{\times}10^4}$ | $\mathbf{2.04{\times}10^4}$ | $1.07{\times}10^3$ | $\mathbf{3.92{\times}10^4}$ | $\mathbf{4.15{\times}10^4}$ | $1.97{\times}10^3$ | $\mathbf{9.54{\times}10^4}$ | $\mathbf{10.0{\times}10^4}$ | $4.40{\times}10^3$ |

| | Tri-FJSP - 5j5m | | | Tri-FJSP - 10j5m | | | Tri-FJSP - 25j5m | | |
|---|---|---|---|---|---|---|---|---|---|
| Method | mean | max | std | mean | max | std | mean | max | std |
| NSGA-II | $2.06{\times}10^6$ | $2.22{\times}10^6$ | $1.32{\times}10^5$ | $5.53{\times}10^6$ | $5.95{\times}10^6$ | $3.09{\times}10^5$ | $2.05{\times}10^7$ | $2.18{\times}10^7$ | $1.13{\times}10^6$ |
| irace | $\mathbf{2.11{\times}10^6}$ | $\mathbf{2.26{\times}10^6}$ | $1.16{\times}10^5$ | $5.47{\times}10^6$ | $5.82{\times}10^6$ | $2.65{\times}10^5$ | $2.07{\times}10^7$ | $2.20{\times}10^7$ | $1.07{\times}10^6$ |
| SMAC3 | $2.09{\times}10^6$ | $2.25{\times}10^6$ | $1.23{\times}10^5$ | $5.65{\times}10^6$ | $6.05{\times}10^6$ | $2.91{\times}10^5$ | $2.07{\times}10^7$ | $2.20{\times}10^7$ | $1.01{\times}10^6$ |
| MADAC | $1.99{\times}10^6$ | $2.14{\times}10^6$ | $8.86{\times}10^4$ | $5.39{\times}10^6$ | $5.87{\times}10^6$ | $5.97{\times}10^5$ | $2.09{\times}10^7$ | $2.20{\times}10^7$ | $1.97{\times}10^6$ |
| GS-MODAC | $2.10{\times}10^6$ | $2.25{\times}10^6$ | $1.16{\times}10^5$ | $\mathbf{5.70{\times}10^6}$ | $\underline{6.09{\times}10^6}$ | $2.99{\times}10^5$ | $\mathbf{2.14{\times}10^7}$ | $\underline{2.27{\times}10^7}$ | $1.09{\times}10^6$ |

| | Penta-FJSP - 5j5m | | | Penta-FJSP - 10j5m | | | Penta-FJSP - 25j5m | | |
|---|---|---|---|---|---|---|---|---|---|
| Method | mean | max | std | mean | max | std | mean | max | std |
| NSGA-II | $6.01{\times}10^{10}$ | $6.48{\times}10^{10}$ | $3.70{\times}10^9$ | $3.96{\times}10^{11}$ | $4.31{\times}10^{11}$ | $2.42{\times}10^{10}$ | $5.08{\times}10^{12}$ | $5.48{\times}10^{12}$ | $2.75{\times}10^{11}$ |
| irace | $6.08{\times}10^{10}$ | $6.49{\times}10^{10}$ | $2.98{\times}10^9$ | $4.03{\times}10^{11}$ | $4.38{\times}10^{11}$ | $2.35{\times}10^{10}$ | $5.18{\times}10^{12}$ | $5.63{\times}10^{12}$ | $2.92{\times}10^{11}$ |
| SMAC3 | $6.08{\times}10^{10}$ | $6.50{\times}10^{10}$ | $3.29{\times}10^9$ | $3.97{\times}10^{11}$ | $4.29{\times}10^{11}$ | $2.24{\times}10^{10}$ | $4.95{\times}10^{12}$ | $5.33{\times}10^{12}$ | $2.71{\times}10^{11}$ |
| MADAC | $5.82{\times}10^{10}$ | $6.28{\times}10^{10}$ | $2.72{\times}10^9$ | $3.91{\times}10^{11}$ | $4.29{\times}10^{11}$ | $4.36{\times}10^{10}$ | $5.1{\times}10^{12}$ | $5.74{\times}10^{12}$ | $5.01{\times}10^{11}$ |
| GS-MODAC | $\mathbf{6.15{\times}10^{10}}$ | $\underline{6.58{\times}10^{10}}$ | $3.40{\times}10^9$ | $\mathbf{4.16{\times}10^{11}}$ | $\mathbf{4.52{\times}10^{11}}$ | $2.40{\times}10^{10}$ | $\mathbf{5.62{\times}10^{12}}$ | $\underline{6.07{\times}10^{12}}$ | $3.20{\times}10^{11}$ |

| | Bi-CVRP - 100 | | | Bi-CVRP - 200 | | | Bi-CVRP - 500 | | |
|---|---|---|---|---|---|---|---|---|---|
| Method | mean | max | std | mean | max | std | mean | max | std |
| NSGA-II | $1.34{\times}10^2$ | $1.47{\times}10^2$ | 7.84 | $1.56{\times}10^2$ | $1.72{\times}10^2$ | 9.05 | $2.27{\times}10^2$ | $2.48{\times}10^2$ | $1.27{\times}10^1$ |
| irace | $1.34{\times}10^2$ | $1.48{\times}10^2$ | 8.02 | $1.57{\times}10^2$ | $1.72{\times}10^2$ | 9.53 | $2.27{\times}10^2$ | $2.48{\times}10^2$ | $1.26{\times}10^1$ |
| SMAC3 | $1.34{\times}10^2$ | $1.46{\times}10^2$ | 7.89 | $1.57{\times}10^2$ | $1.73{\times}10^2$ | 9.59 | $2.27{\times}10^2$ | $2.51{\times}10^2$ | $1.42{\times}10^1$ |
| MADAC | $\mathbf{1.35{\times}10^2}$ | $\mathbf{1.49{\times}10^2}$ | 8.01 | $\mathbf{1.61{\times}10^2}$ | $\mathbf{1.76{\times}10^2}$ | 9.22 | $2.33{\times}10^2$ | $2.54{\times}10^2$ | $1.29{\times}10^1$ |
| GS-MODAC | $\mathbf{1.35{\times}10^2}$ | $1.48{\times}10^2$ | 7.95 | $1.60{\times}10^2$ | $\mathbf{1.76{\times}10^2}$ | 9.50 | $\mathbf{2.35{\times}10^2}$ | $\mathbf{2.59{\times}10^2}$ | $1.41{\times}10^1$ |

**Testing.** After training, the GS-MODAC agent is ready to be applied to tune the parameters of NSGA-II to solve unseen problem instances. Each experiment is performed by running each algorithm 10 times on 100 test instances for comparison. The evaluation is based on three metrics: average hypervolume (mean), best hypervolume (max), and standard deviation (std), which are computed by averaging all test instances for each problem. Hypervolumes are calculated using predefined reference points for each instance to ensure a fair comparison. The paper highlights the highest mean and max hypervolumes in bold and underlined values that significantly outperform all other methods using the Wilcoxon rank-sum test ($p < 0.05$).

## 4.1. Experimental results

We have formulated research questions to evaluate the performance of the proposed GS-MODAC method. Specifically, these questions assess the effectiveness of GS-MODAC in comparison to existing methods, its ability to generalize to previously unseen instances of varying sizes, its adaptability to more complex problem variants, and its scalability across different objectives.

**RQ1: How does GS-MODAC perform compared to the base algorithm NSGA-II and three AC baseline methods for various problem types and sizes of objectives?** Table 1 presents the performances of various methods, including the mean average performance, mean best-found solution, and the standard deviations for each method on two different problem types. The results highlight the effectiveness of GS-MODAC in controlling evolutionary parameters, achieving the best average and best-found solutions. For the smallest instance size in two-objective problems (Bi-), the baseline methods perform competitively, with the MADAC and irace configured baselines finding comparable mean and max solutions. However, GS-MODAC consistently excels in problem configurations with larger objective spaces, such

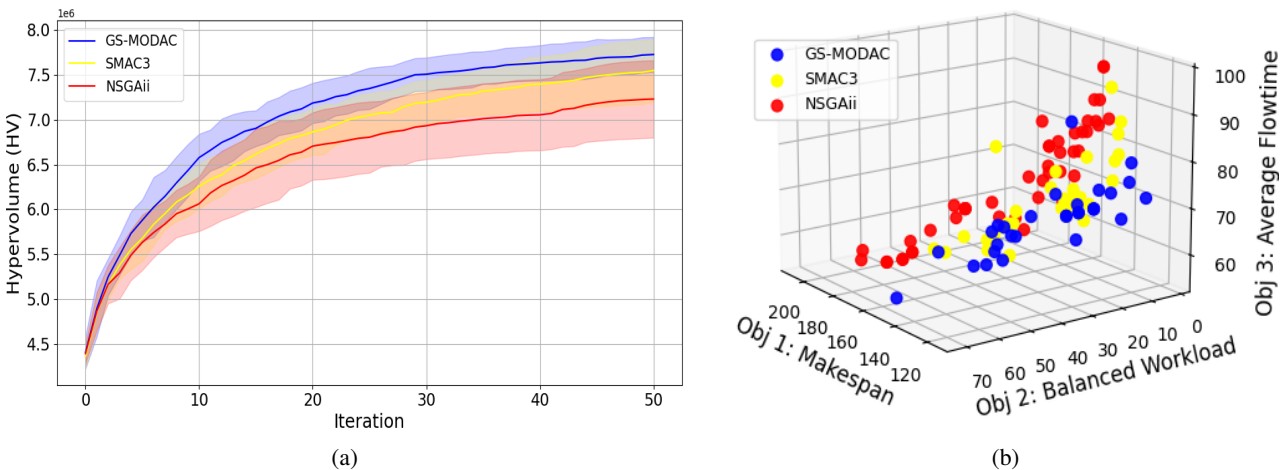

(a)                                            (b)

Figure 2: Comparison of GS-MODAC, SMAC3, and NSGA-II solution methods: (a) Average convergence rates and (b) Pareto front distributions.

Table 2: Generalizability of the trained models to solve unseen instances of different sizes.

| Method | Bi-CVRP - 100 | | | Bi-CVRP - 200 | | | Bi-CVRP - 500 | | |
|---|---|---|---|---|---|---|---|---|---|
| | mean | max | std | mean | max | std | mean | max | std |
| NSGA-II | $1.34 \times 10^2$ | $1.47 \times 10^2$ | 7.84 | $1.56 \times 10^2$ | $1.72 \times 10^2$ | 9.05 | $2.27 \times 10^2$ | $2.48 \times 10^2$ | $1.27 \times 10^1$ |
| GS-MODAC - 100 | $\mathbf{1.35 \times 10^2}$ | $\mathbf{1.48 \times 10^2}$ | 7.95 | $1.59 \times 10^2$ | $1.75 \times 10^2$ | 9.25 | $2.32 \times 10^2$ | $2.55 \times 10^2$ | $1.31 \times 10^1$ |
| GS-MODAC - 200 | $\mathbf{1.35 \times 10^2}$ | $\mathbf{1.48 \times 10^2}$ | 8.22 | $\mathbf{1.60 \times 10^2}$ | $\mathbf{1.76 \times 10^2}$ | 9.50 | $2.33 \times 10^2$ | $2.56 \times 10^2$ | $1.36 \times 10^1$ |
| GS-MODAC - 500 | $1.33 \times 10^2$ | $1.47 \times 10^2$ | 8.52 | $\mathbf{1.60 \times 10^2}$ | $1.75 \times 10^2$ | 9.33 | $\mathbf{2.35 \times 10^2}$ | $\mathbf{2.59 \times 10^2}$ | $1.41 \times 10^1$ |
| GS-MODAC - all sizes | $1.34 \times 10^2$ | $\mathbf{1.48 \times 10^2}$ | 7.97 | $1.59 \times 10^2$ | $1.74 \times 10^2$ | 9.03 | $2.33 \times 10^2$ | $\mathbf{2.59 \times 10^2}$ | $1.41 \times 10^1$ |

as problems with more objectives and larger combinatorial search spaces (large instances configurations). This is particularly evident in the FJSP problem configurations with five objectives (Penta-), where GS-MODAC finds significantly better solutions than all baselines regarding mean and max found solutions. Specifically, for the Penta-FJSP problem configurations with 25 jobs and 5 machines, GS-MODAC's mean and maximum solutions are 8.2% and 5.7% better, respectively, than the best-performing baselines (irace and MADAC) and 10.6% and 10.8% better than the vanilla configured NSGA-II method.

Figures 2a and 2b illustrate that the GS-MODAC method converges significantly faster in finding the best hypervolume for a Tri-FJSP with 10 jobs and 5 machines, compared to baseline approaches. It achieves a better-converged hypervolume, reaching lower minimum values for each objective, and shows a wider spread across the different objective axes. Similar convergence patterns were observed for other instances, demonstrating the robustness of GS-MODAC. In Appendix D, we provide further analysis using alternative performance metrics to demonstrate GS-MODAC's ability to converge towards the true Pareto front while maintaining a diverse and high-quality set of solutions.

**RQ2: How well do the trained GS-MODAC models generalize to previously unseen instances of varying sizes?** We assess the ability of the trained GS-MODAC models to solve previously unseen instances of different sizes. The results of this evaluation are presented in Table 2. In the table, the rows show the instance sizes used for training, while the columns show the instance sizes on which trained models are tested. We found that models trained on smaller problem instances and deployed on larger instances experienced a slight decline in performance, but still managed to achieve performance comparable to the best performing baseline method (MADAC) while outperforming the other baselines. Moreover, models trained on a more diverse set of instance sizes can effectively learn a robust, well-performing, all-around policy. The results suggest that our models could generalize and solve problem instances beyond the size on which they were trained.

**RQ3: How effectively can trained GS-MODAC models handle previously unseen, more complex problem variants?** We evaluate the generalization capability of the trained GS-MODAC models by applying them to previously unseen instances of a different and more complex prob-

Table 3: Comparing the generalizability of trained models to solve instances of more complex problem variants.

| Method | Bi-DAFJS-SDST | | | Tri-DAFJS-SDST | | | Penta-DAFJS-SDST | | |
|---|---|---|---|---|---|---|---|---|---|
| | mean | max | std | mean | max | std | mean | max | std |
| NSGA-II | $1.32\times10^6$ | $1.41\times10^6$ | $7.93\times10^4$ | $3.46\times10^8$ | $3.75\times10^8$ | $2.01\times10^7$ | $8.19\times10^{14}$ | $8.92\times10^{14}$ | $4.89\times10^{13}$ |
| irace | $1.41\times10^6$ | $1.47\times10^6$ | $5.60\times10^4$ | $3.49\times10^8$ | $3.75\times10^8$ | $1.86\times10^7$ | $8.10\times10^{14}$ | $8.76\times10^{14}$ | $3.97\times10^{13}$ |
| SMAC3 | $1.40\times10^6$ | $1.48\times10^6$ | $7.08\times10^4$ | $3.37\times10^8$ | $3.64\times10^8$ | $1.94\times10^7$ | $8.26\times10^{14}$ | $9.01\times10^{14}$ | $4.76\times10^{13}$ |
| GS-MODAC - FJSP-10j5m | $1.42\times10^6$ | $1.50\times10^6$ | $6.30\times10^4$ | $3.68\times10^8$ | $3.94\times10^8$ | $2.02\times10^7$ | $\mathbf{9.11\times10^{14}}$ | $9.93\times10^{14}$ | $5.48\times10^{13}$ |
| GS-MODAC - DAFJS-SDST | $\mathbf{1.43\times10^6}$ | $\mathbf{1.51\times10^6}$ | $6.48\times10^4$ | $\mathbf{3.73\times10^8}$ | $\mathbf{4.00\times10^8}$ | $2.25\times10^7$ | $9.05\times10^{14}$ | $\mathbf{1.00\times10^{15}}$ | $6.45\times10^{13}$ |

Table 4: Comparing the generalizability of the trained models to solve problem configuration to optimize different objectives that were not optimized in training.

| Method | Bi-FJSP* - 5j5m | | | Bi-FJSP* - 10j5m | | | Bi-FJSP* - 25j5m | | |
|---|---|---|---|---|---|---|---|---|---|
| | mean | max | std | mean | max | std | mean | max | std |
| NSGA-II | $3.49\times10^3$ | $3.57\times10^3$ | $4.05\times10^1$ | $9.64\times10^3$ | $9.88\times10^3$ | $1.31\times10^2$ | $\mathbf{3.93\times10^4}$ | $\mathbf{3.98\times10^4}$ | $2.57\times10^2$ |
| irace | $3.50\times10^3$ | $3.58\times10^3$ | $4.53\times10^1$ | $\mathbf{9.66\times10^3}$ | $9.91\times10^3$ | $1.48\times10^2$ | $3.92\times10^4$ | $3.97\times10^4$ | $2.62\times10^2$ |
| SMAC3 | $3.50\times10^3$ | $3.58\times10^3$ | $4.60\times10^1$ | $9.61\times10^3$ | $9.87\times10^3$ | $1.58\times10^2$ | $3.92\times10^4$ | $3.97\times10^4$ | $3.24\times10^2$ |
| GS-MODAC | $\mathbf{3.51\times10^3}$ | $\mathbf{3.58\times10^3}$ | $3.97\times10^1$ | $\mathbf{9.66\times10^3}$ | $\mathbf{9.93\times10^3}$ | $1.60\times10^2$ | $\mathbf{3.93\times10^4}$ | $\mathbf{3.98\times10^4}$ | $2.42\times10^2$ |

lem variant. This problem extends the Bi-, Tri- and Penta-objective FJSP with assembly constraints and sequence-dependent setup times, including additional precedence constraints between different jobs and setup times operations on machines subject to the scheduling sequence. We tested the proposed method on a so-called 'DAFJS' scheduling problem as provided in Birgin et al. (2014), which has been extended with sequence-dependent setup times. The results, shown in Table 3, indicate that GS-MODAC trained on DAFJS-SDST demonstrates superior performance in most cases, except for the mean HV in the Penta-objectives variant. Furthermore, the model configuration trained on the 10j5m problem variants effectively transfers to more complex problem scenarios. In particular, GS-MODAC trained in 10j5m configurations outperforms all other baselines specifically tailored to the DAFJS problem variant.

**RQ4: How effectively does the GS-MODAC model trained on a specific set of objectives adapt to different objectives than those encountered during training?** We evaluate the ability of trained models to generalize across different variants of FJSP problems configured to optimize for objectives different from those explored in training. Specifically, we assess how well models trained to optimize objectives A and B perform when applied to a variant of the problem that is instead configured to optimize for objectives C and D. This transfer scenario is denoted as Bi-FJSP*. From Table 4, it is clear that trained models can be transferred to other problem configurations, finding solutions of similar or better quality than the configured baselines, with a similar performance gap as consistently observed for two objective problem variants displayed in Table 1.

## 5. Conclusion

This paper introduces Graph-Supported Multi-Objective Dynamic Algorithm Configuration (GS-MODAC), a novel approach that integrates Graph Neural Networks (GNNs) and Deep Reinforcement Learning (DRL) to dynamically configure Evolutionary Algorithms for solving multi-objective combinatorial optimization problems. GS-MODAC represents the evolving state of the search process as a graph, capturing the structural and convergence dynamics of solutions across multiple objectives. To ensure robustness across different problem instances and problem configurations, we propose an instance-agnostic reward function that is suitable to diverse problem types and sizes. Empirical results demonstrate that GS-MODAC outperforms both traditional Algorithm Configuration approaches and state-of-the-art DRL-based dynamic algorithm, achieving better effectiveness and adaptability. Additionally, our method generalizes effectively to larger, more constrained problem instances not encountered during training.

## Acknowledgements

This work is supported by the Luxembourg National Research Fund (FNR) (15706426).

## Impact Statement

This paper presents work whose goal is to advance the field of Machine Learning. There are many potential societal consequences of our work, none which we feel must be specifically highlighted here.

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

## A. Test problem configurations

The **Flexible Job Shop Scheduling Problem (FJSP)** is a popular scheduling problem where multiple jobs, each composed of several operations that must be completed in a specific order, must be scheduled to a set of machines. The problem contains a set of $n$ independent jobs $J = \{J_1, J_2, \ldots, J_n\}$ and $m$ independent machines $M = \{M_1, M_2, \ldots, M_m\}$, which together for an $n \times m$ Flexible Job Shop Scheduling Problem (FJSP). Each job $J_i$ consists operations $O_{i,j}$, where $O_{i,j}$ represents the $j$-th operation of the $i$-th job. These operations must be executed in sequence, meaning $O_{i,j+1}$ may only start after $O_{i,j}$ is completed. The processing time for operation $O_{i,j}$ on machine $M_k$ is denoted as $t_{i,j,k}$ and is known in advance. Each machine $M_k$ can process only one operation at a time, and operations cannot be interrupted (no preemption). The start and completion times for operation $O_{i,j}$ are denoted as $S_{i,j}$ and $C_{i,j}$ respectively, while $O_k$ is the set of operations assigned to on machine $M_k$.

This work focuses on five key minimization objectives commonly used in scheduling:

- Makespan: The total time required to complete all jobs, represented as $C_{\max} = \max_{i=1,\ldots,n} C_{i,j}$.

- Balance Workload: The disparity in workload distribution across machines, represented as $W_{\mathrm{bal}} = W_{\max} - W_{\min}$, where $W_{\min} = \min_{k=1,\ldots,m} \sum_{(i,j) \in O_k} t_{i,j,k}$.

- Average flowtime: The average time duration jobs take from start to completion $F_{\mathrm{avg}} = \frac{1}{n} \sum_{i=1}^{n} (C_{i,\mathrm{last}} - S_{i,\mathrm{first}})$.

- Total Workload: The cumulative sum of processing times for all jobs, defined as $W_{\mathrm{total}} = \sum_{k=1}^{m} \sum_{(i,j) \in O_k} t_{i,j,k}$.

- Maximum flowtime: Denoting the longest time any job spends in the system from start to completion, defined as $F_{\max} = \max_{i=1,\ldots,n} (C_{i,\mathrm{last}} - S_{i,\mathrm{first}})$.

The **Capacitated Vehicle Routing Problem (CVRP)** is concerned with a fleet of vehicles that must deliver goods from a central depot to a set of customer locations while satisfying capacity constraints. The problem contains a set of $n$ customer locations $C = \{C_1, C_2, \ldots, C_n\}$, a depot location $C_0$, and $m$ identical vehicles. Each location $C_i$ has a demand $q_i$ representing the quantity of goods that need to be delivered to that particular customer. Each vehicle has a capacity of $Q$, representing the maximum total demand it can serve in a single route. Each vehicle $k$ can serve a demand of $\sum_{i=1}^{n} q_i \times y_{ik} \leq Q$, where $y_{ik}$ is a binary decision variable indicating whether vehicle $k$ serves customer $i$. The distance matrix $D$ is defined as $d_{ij}$, containing the distances between all pairs of locations, including customer locations and the depot, encapsulating the travel costs or distances associated with moving from one location to another.

The objectives considered in this work are to minimize the total distance traveled by all vehicles and the longest route:

- Total Travel Distance: $D_{\mathrm{total}} = \sum_{k=1}^{m} \sum_{i=1}^{n} \sum_{j=1}^{n} d_{ij} \times x_{ijk}$

- Longest Route: $D_{\max} = \max_{k=1}^{m} \sum_{i=1}^{n} \sum_{j=1}^{n} d_{ij} \times x_{ijk}$

## B. Multi-Objective Algorithms for MOCO

**NSGA-II for FJSP.** To assess the efficacy of the proposed approach for FJSP, we devise a multi-objective Genetic Algorithm (GA) formulation inspired by Zhang et al. (2011) and following the implementation of (Reijnen et al., 2023a). The solutions entail two integral components: Machine Selection and Operation Sequence. The first allocates operations to machines, while the second establishes the precedence of operations on the designated machines. Illustrated in Figure 3, a value of '4' in the initial position of Machine Selection indicates the scheduling of operation $O_{1,1}$ on the fourth machine alternative. Subsequently, the Operation Sequence component arranges this operation as second, after $O_{2,1}$.

The population is initialized using Global, Local, and Random Methods. Global Method assigns operations to machines sequentially, minimizing the total processing times of individual machines. Local Method minimizes the max machine processing times for individual jobs. Random Method allocates operations to machines randomly. The Operation Sequence is initialized randomly for all methods. Crossover is applied to the Machine Selection component using two-point and uniform crossover while precedence-preserving order-based crossover (POX) is applied to the Operation Sequence component. POX preserves relative scheduling positions for a randomly selected set of jobs and reschedules the remaining operations according to the other crossovered individual solution. We generate 60% of the initial population using Global Method,

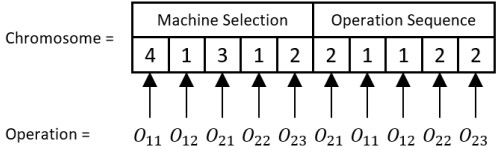

Figure 3: Chromosome Representation FJSP MOGA (Zhang et al., 2011)

30% using Local Method, and 10% using Random Method. Machine Selection crossovers are in 50% two-point and 50% uniform crossover. To solve the multi-objective FJSP variant using the GA formulation from Zhang et al. (2011), we employ Non-dominated Sorting Genetic Algorithm-II (NSGA-II) for selection (Deb et al., 2002).

**NSGA-II for CVRP.** Subsequently, we apply a multi-objective Genetic Algorithm (GA) formulation to assess the efficacy of the proposed approach for CVRP. The solutions are initialized with random routes, where each solution is represented as a list of values corresponding to the sequence in which customers are visited in the CVRP.

The selected parents undergo crossover and mutation to produce offspring, using ordered crossover and a shuffle mutation; crossing over two segments from two selected parent solutions, and randomly swapping elements within solutions with a given probability. The next generation is formed by selecting individuals from the combined population based on their rank (front) and crowding distance. The algorithm prioritizes individuals from lower fronts and those with higher crowding distances to ensure a diverse and high-quality population. Non-dominated Sorting Genetic Algorithm-II (NSGA-II) is applied for the selection (Deb et al., 2002).

**MOPSO for CVRP.** We define a Multi-Objective Particle Swarm Optimization (MOPSO) algorithm for the Capacitated Vehicle Routing Problem (CVRP). In this algorithm, solutions (particles) are initialized with random routes, represented as a list of random values where each value corresponds to a customer in the CVRP. Each particle also has associated velocities that represent changes in these routes. The initial fitness values for each particle are calculated by sorting the customers based on the values in the particle's position to determine the routes. Each particle's personal best solution is recorded, and all the best-found solutions are stored in a separate list.

In each generation of the search, the positions and velocities of the particles are updated based on their personal best and a global best chosen from the Pareto front (randomly selected when multiple best solutions are available). The velocity update formula incorporates cognitive coefficients ($\phi 1$), social coefficients ($\phi 2$), and an inertia weight. Initially, random coefficients (u1 and u2) are generated for each particle dimension to balance exploration and exploitation. The velocity update consists of two components: one influenced by the particle's personal best and the other by the global best from the Pareto front. The velocity for each particle dimension is calculated using these components, scaled by the respective random coefficients and adjusted by the inertia weight. The updated velocity is clamped within predefined minimum (min) and maximum (max) bounds to remain within valid bounds. The particle's new position is determined by adding the updated velocity to the current position. Finally, each position is clamped to remain within valid bounds, typically between 0 and 1, ensuring the particle stays within the feasible solution space.

After the update, the fitness of the particles is evaluated. The particles' personal bests and the list of best solutions are updated using non-dominated sorting to retrieve Pareto-optimal solutions. A selection mechanism based on Pareto dominance (using NSGA-II) is applied to maintain a diverse and optimal set of solutions in the population. For this work, we configure the vanilla MOPSO algorithm for CVRP with the following parameters: social and cognitive coefficients are configured as 2.0, an inertia weight of 0.9, and we use a population size of 50 particles for 50 generations. GS-MODAC is configured to tune the social and cognitive coefficients between 1 and 3 and the inertia weight factor between 0.6 and 0.9.

## C. Alternative Performance Metrics

We further evaluate performances using additional metrics commonly employed in multi-objective optimization research: Inverted Generational Distance (IGD), Inverted Generational Distance Plus (IGD+), and the number of non-dominated solutions. These results are gathered using the same setup as used in the paper for the J25m5 scheduling problem with 2,3 and 5 objectives. The results, shown in Table 5, highlight the effectiveness of GS-MODAC, as it finds a significantly higher number of "best" solutions (max) and achieves lower IGD+ values.

In terms of IGD, GS-MODAC outperforms the baseline methods in the experiments with more objectives. It is important to note that while IGD provides valuable insights into the proximity of solutions to the Pareto front, it is sensitive to the distribution of solutions and is more subject to outliers. In contrast, IGD+ is less sensitive to these factors, making it a more reliable measure to evaluate the overall quality and diversity of solutions. Therefore, the consistently lower IGD+ values across multiple objectives achieved by GS-MODAC highlight its ability to converge to the true Pareto front while maintaining a diverse set of high-quality solutions.

Table 5: Additional Performance Metrics: Inverted Generational Distance (IGD), Inverted Generation Distance Plus (IGD+), and nr. of non-dominated solutions.

| | Bi-FJSP - 25j5m | | | | | | | | |
| | IGD | | | IGD+ | | | non-dominated solutions | | |
| | mean | min | std | mean | min | std | mean | max | std |
|---|---|---|---|---|---|---|---|---|---|
| NSGA-II | 19.07 | 8.47 | 11.49 | 15.79 | 4.05 | 12.25 | 5.06 | 8.78 | 2.11 |
| irace | 15.66 | 7.23 | 8.41 | 11.23 | 2.52 | 9.10 | 4.92 | 8.07 | **1.94** |
| SMAC3 | **15.13** | **6.99** | **7.75** | 15.13 | 6.99 | **7.75** | 4.99 | 8.26 | 1.98 |
| GS-MODAC | 15.77 | 7.41 | 9.10 | **9.82** | **1.25** | 10.10 | **6.81** | **20.69** | 5.80 |
| | Tri-FJSP - 25j5m | | | | | | | | |
| | IGD | | | IGD+ | | | non-dominated solutions | | |
| | mean | min | std | mean | min | std | mean | max | std |
| NSGA-II | 22.20 | 15.77 | **6.00** | 18.09 | 10.11 | **6.78** | 36.29 | 54.10 | 10.18 |
| irace | 19.23 | 13.44 | 6.09 | 12.29 | 5.36 | 6.83 | 35.30 | 51.29 | 10.06 |
| SMAC3 | **19.19** | **13.07** | 6.13 | 11.24 | 3.83 | 7.02 | 35.25 | 52.95 | 9.81 |
| GS-MODAC | 20.63 | 13.86 | 6.77 | **8.40** | **2.46** | 6.79 | **36.64** | **54.83** | **11.07** |
| | Penta-FJSP - 25j5m | | | | | | | | |
| | IGD | | | IGD+ | | | non-dominated solutions | | |
| | mean | min | std | mean | min | std | mean | max | std |
| NSGA-II | 28.58 | 23.42 | 4.19 | 21.18 | 14.18 | **4.70** | 172.20 | 223.14 | 32.46 |
| irace | 23.97 | 19.88 | **4.09** | 13.67 | 7.38 | 4.78 | 203.94 | 263.34 | 40.22 |
| SMAC3 | 26.10 | 21.16 | 4.46 | 17.17 | 9.91 | 5.22 | 185.93 | 243.54 | **35.94** |
| GS-MODAC | **23.82** | **19.08** | 5.06 | **8.71** | **3.25** | 5.03 | **231.13** | **311.21** | 52.14 |

## D. Alternative MOEA results

We show another instantiation of the proposed GS-MODAC method, where the DRL agent dynamically configures the parameters of a multi-objective PSO (MOPSO) algorithm. Table 6 shows GS-MODAC can effectively improve the performance of MOPSO, achieving better results on solving the two-objective CVRP problems with sizes 20, 50, and 100.

Table 6: Performance comparison of the proposed method for dynamic algorithm configuration of Multi-Objective Particle Swarm Optimization (MOPSO) Algorithm.

| | Bi-CVRP - 20 | | | Bi-CVRP - 50 | | | Bi-CVRP - 100 | | |
| Method | mean | max | std | mean | max | std | mean | max | std |
|---|---|---|---|---|---|---|---|---|---|
| MOPSO | $3.21 \times 10^1$ | $3.75 \times 10^1$ | 3.47 | $5.82 \times 10^1$ | $6.90 \times 10^1$ | **5.65** | $8.67 \times 10^1$ | $9.75 \times 10^1$ | **5.87** |
| GS-MODAC | $\mathbf{3.28 \times 10^1}$ | $\mathbf{3.77 \times 10^1}$ | **3.20** | $\mathbf{6.27 \times 10^1}$ | $\mathbf{8.08 \times 10^1}$ | 9.73 | $\mathbf{1.06 \times 10^2}$ | $\mathbf{1.46 \times 10^2}$ | $2.42 \times 10^1$ |

## E. Ablation study

An ablation study was conducted to account for the performance of the different components of the proposed method. As a first ablation, we trained GS-MODAC without the additional feature vector that contains the normalized remaining search budget. Table 7 shows that, without this vector, the performance of the proposed method decreased on average with 0.8%, 3.2%, and 1.7%, respectively, for 2, 3, and 5 objectives to solve the scheduling problem with 25j5m instances. Another ablation was conducted with only one GCN layer. This resulted in an average performance decrease of 1.7%, 3.2%, and 3.4% for 2, 3, and 5 objectives.

In addition, we adapt GS-MODAC for the Penta-FJSP - 25j5m problem by replacing the GCN layers with Transformers and Graph Attention Networks (GAT). The results presented in Table 8 indicate that Transformers are a viable alternative, with

Table 7: Ablation study, comparing GS-MODAC configured without additional feature vector and with one configured GCN layer.

| | Bi-FJSP - 25j5m | | | Tri-FJSP - 25j5m | | | Penta-FJSP - 25j5m | | |
|---|---|---|---|---|---|---|---|---|---|
| | mean | max | std | mean | max | std | mean | max | std |
| MADAC | $9.24{\times}10^4$ | $9.72{\times}10^4$ | $\mathbf{3.09{\times}10^3}$ | $2.09{\times}10^7$ | $2.20{\times}10^7$ | $1.97{\times}10^6$ | $5.1{\times}10^{12}$ | $5.74{\times}10^{12}$ | $5.01{\times}10^{11}$ |
| GS-MODAC (No feature) | $9.47{\times}10^4$ | $9.92{\times}10^4$ | $4.21{\times}10^3$ | $2.07{\times}10^7$ | $2.21{\times}10^7$ | $1.10{\times}10^6$ | $5.53{\times}10^{12}$ | $6.02{\times}10^{12}$ | $3.42{\times}10^{11}$ |
| GS-MODAC (One GCN) | $9.38{\times}10^4$ | $9.88{\times}10^4$ | $4.87{\times}10^3$ | $2.07{\times}10^7$ | $2.19{\times}10^7$ | $\mathbf{1.02{\times}10^6}$ | $5.49{\times}10^{12}$ | $5.98{\times}10^{12}$ | $3.37{\times}10^{11}$ |
| GS-MODAC | $\mathbf{9.54{\times}10^4}$ | $\mathbf{10.0{\times}10^4}$ | $4.40{\times}10^3$ | $\mathbf{2.14{\times}10^7}$ | $\mathbf{2.27{\times}10^7}$ | $1.09{\times}10^6$ | $\mathbf{5.62{\times}10^{12}}$ | $\mathbf{6.07{\times}10^{12}}$ | $\mathbf{3.20{\times}10^{11}}$ |

an average performance of only 0.3% lower than GCN and its best-found solutions only 0.5% worse. The performance difference of GS-MODAC configured with GAT layers is more substantial, with an average degradation of 1.4%.

Table 8: Comparison of different network architectures (GCN, Transformer, GAT) for GS-MODAC.

| | Penta-FJSP - 25j5m | | |
|---|---|---|---|
| | mean | max | std |
| GCN | $\mathbf{5.62{\times}10^{12}}$ | $\mathbf{6.07{\times}10^{12}}$ | $\mathbf{3.20{\times}10^{11}}$ |
| transformer | $5.61{\times}10^{12}$ | $6.04{\times}10^{12}$ | $3.60{\times}10^{11}$ |
| GAT | $5.54{\times}10^{12}$ | $6.04{\times}10^{12}$ | $3.35{\times}10^{11}$ |

# F. Complexity Analysis

We profiled GS-MODAC to assess its computational complexity, focusing on graph state configuration and policy network inference. The results show that the actor's inference time is 0.13 seconds, and the state extraction takes 0.2 seconds, together accounting for 2.0% of the total time for the smallest scheduling problem instances. For larger problems, this proportion decreases significantly as solution evaluations dominate the computation. Despite a slight overhead, its substantial performance gains justify the minimal additional cost of GS-MODAC.

Table 9: Breakdown of GS-MODAC's computational components, highlighting total inference time, state configuration, and policy inference time.

| | Bi-FJSP - 5j5m | Penta-FJSP - 5j5m | Bi-FJSP - 25j5m | Penta-FJSP - 25j5m |
|---|---|---|---|---|
| Total Inference Time | 15.09s | 15.46s | 305s | 302s |
| Total State Configuration Time | 0.18s | 0.21s | 0.23s | 0.22s |
| Total Policy Inference Time | 0.12s | 0.12s | 0.14s | 0.13s |

# G. Comparison to End-to-End method P-MOCO

We compare GS-MODAC with P-MOCO (Lin et al., 2022), a commonly used learning-based approach for Pareto set learning. It is important to note that P-MOCO features a specialized network structure for simple TSP and CVRP and cannot solve scheduling problems (such as FJSP), nor work for alternative, non-distance-based objectives. Hence, we compare both methods to solve the CVRP problem with 2 objectives. We train P-MOCO according to the details provided in Lin et al. (2022) and train both methods on the same set of instances of size 100. We evaluate its performance based on the setup described in Section 4, utilizing the same instances and reference points recorded in the publication. The results, presented in Table 10, compare the best obtained HV values, aligned with the experimental setup of Lin et al. (2022).

Table 10: Comparison of Hypervolume (HV) values achieved by NSGA-II, and by P-MOCO and GS-MODAC (trained on size 100) for Bi-CVRP instances of varying sizes.

| | Bi-CVRP - 20 | Bi-CVRP - 50 | Bi-CVRP - 100 |
|---|---|---|---|
| NSGA-II | 42.86 | 151.24 | 363.87 |
| P-MOCO | 34.71 | 152.83 | 438.06 |
| GS-MODAC | **45.18** | 152.87 | 366.63 |

The results indicate that P-MOCO performs better than GS-MODAC when trained and tested on instances of size 100, which is expected since P-MOCO learns policies tailored to specific instances. However, in terms of generalizability, P-MOCO is inferior to GS-MODAC, as seen in the performances on Bi-CVRP20 and Bi-CVRP50. This indicates that GS-MODAC has significantly better generalization capability than P-MOCO, which is somewhat overfitted to a specific size used in training. Additionally, we also observe that GS-MODAC outperforms NSGA-II when generalizing to different sizes.

To further assess robustness, we conducted an additional comparison between GS-MODAC and P-MOCO. Both methods were trained on problem instances of size 100, and tested on problem instances generated according to a normal distribution with a mean of 0.3 and a standard deviation of 0.1, with 5% outliers (note: training instances are generated with uniform distributions). The results demonstrate that GS-MODAC consistently outperforms P-MOCO across all sizes, indicating its superior generalization capability. Unlike P-MOCO, which tends to overfit not only to a specific problem size but also to the distribution of training instances, GS-MODAC shows robust performance across different instance distributions. Additionally, GS-MODAC keeps surpassing NSGA-II in the generalization to various instance distributions and sizes.

Table 11: Performances in terms of Hypervolume (HV) on Bi-CVRP instances with different distributions and outliers, compared to the training instances.

|  | Bi-CVRP - 20 | Bi-CVRP - 50 | Bi-CVRP - 100 |
| --- | --- | --- | --- |
| NSGA-II | 59.34 | 192.64 | 455.80 |
| P-MOCO | 51.05 | 186.35 | 454.76 |
| GS-MODAC | **59.55** | **194.47** | 458.46 |

