# OpenReview forum: "Graph-Supported Dynamic Algorithm Configuration for Multi-Objective Combinatorial Optimization"
_ICML.cc/2025/Conference — ICML 2025 poster_

### Official Review · Reviewer_qYda · 2025-03-11

**Overall Recommendation:** 4

**Summary:**

The work proposes an extension to the dynamic algorithm configuration (DAC) framework. Particularly, the work proposes to use graph convolutional neural networks to learn embeddings of Pareto fronts such that multi-objective combinatorial optimization algorithms are dynamically optimized while proposing novel solution candidates. The work evaluates the proposed GS-MODAC approach on two different problem domains with various different problem sets and shows that the approach is capable of outperforming classical, static algorithm configuration as well as a multi-agent based dynamic algorithm configuration approach.

## update after rebuttal
I have read all reviews and am keeping my score.

**Claims And Evidence:**

The claimed contributions are mostly convincing and seem well supported by the empirical evidence provided in the main part of the paper and are further complemented by analysis and experiments in the appendix.
Some wordings seem questionable to me.
* Line 025-031 (right side) the statement reads as if "general DAC" approaches are primarily limited to multi-objective continuous optimization. However, this is not the case as other works show that RL-based DAC can be applied in a variety of problem domains (see, e.g. the works by Speck et al. [2021](https://arxiv.org/abs/2006.08246) or [2022](https://andrebiedenkapp.github.io/assets/pdf/paper/22-PRL-DAC4AIPlanning.pdf) highlight that DAC can theoretically outperform the best static configurator and selector while providing additional empirical evidence that is capable of beating a static oracle).
* Line 101-104 (right side) "Unlike static configurations, DAC aims to balance exploration and exploitatoin, increasing the likelihood of finding high-quality solutions." This statement is only partially true. A static configuration can contain hyperparameters of a schedule that influences how the underlying algorithm behaves. A simple example of this are the exponential decay of learning-rates in DL which are often configured via static configurations but dynamically influence the learning behaviour of the target task.

The claim of the reward function being instance-invariant does not seem well supported to me. While I agree that the reward scale is instance-invariant due to the used normalization method, the reward space as such is still instance dependent.
Lastly, from the text it is not clear to me what the optimization objectives of SMAC and irace are. I believe that it is necessary to have a statement to show that SMAC and irace opimize the same normalized objective to highlight that the GS-MODAC approach truly outperforms such strong static baselines.

A small-ish nitpick about the claims: The work discusses generalization to unseen instances, though they are always part of the same domain. Thus the claims about generalization might be reframed from this viewpoint. There can be made no claims about cross-domain generalization, though the work definitely shows that the learned policies can handle a diverse set of problems from the same domain.

**Essential References Not Discussed:**

The works by Speck et al. [2021](https://arxiv.org/abs/2006.08246) & [2022](https://andrebiedenkapp.github.io/assets/pdf/paper/22-PRL-DAC4AIPlanning.pdf) that I also referred to earlier similarly showed that domain-dependent policies (i.e. DAC policies learned in a similar fashion to the work under review) can generalize fairly well to larger problem instances, though with some performance drop-off. Relatedly, Shala et al. [2020](https://ml.informatik.uni-freiburg.de/wp-content/uploads/papers/20-PPSN-LTO-CMA.pdf) showed that DAC policies can generalize to longer horizons, and larger problem dimensions, where the former seemed to be the more limiting factor. Further, the method used by Shala (i.e. Guided Policy Search) was specifically selected to be reward-scale invariant in-order to facilitate better generalizability (see the discussion in the appendix). Lastly, a work by Bordne et al. ([2024](https://arxiv.org/abs/2407.05789)) discussed how to exploit structure in the configuraiton space to facilitate better DAC policies. I believe this idea of exploring ways of exploting structure in DAC problems is related to the idea presented in this work, where structure in the state-space is exploited for better DAC policy learning.

**Experimental Designs Or Analyses:**

See my remarks in "Claims And Evidence".
Additionally, in RQ4 I do not understand the experimental setup and what is actually being investigated. Does the "GS-MODAC model" refer to the learned policy or the graph convolutional neural network. More explanation is needed to understand what we can learn from this experiment and what the results in Table 4 actually represent.

**Methods And Evaluation Criteria:**

The proposed methods and evaluation criteria make sense for the application at hand. Since, to the best of my knowledge, the standard DACBenchmark does not have any relevant benchmarks, it would be good to see that the authors make their benchmark publicly available (potentially as part of DACBench) to facilitate future reproducibility and additional work in this direction.

Another small nitpick: A random configuration policy as baseline would have helped to highlight how meaningful dynamic configuration actually is in the proposed problem setting as the work otherwise does not provide insights on the learned configuration policies.

**Other Comments Or Suggestions:**

N/A

**Other Strengths And Weaknesses:**

The work provides a very elegant solution to the DAC problem and should be of high interest for the AutoML community, but also the RL community as the proposed setting and how to deal with the state space could inform research directions in this community as well.

A weakness is that the work presents itself as being only applicable to multi-objective combinatorial optimization problems. I believe this is not the case. In it's current form the GS-DAC approach should be applicable to a wide variety of mutli-objective problems. There would not even be need for big changes in the algorithm, just the action space for PPO might change. Further, the GS-DAC approach also seems to not be limited to the multi-objective case. While it is highly applicable for this setting, the idea of using graph (convolutional) neural networks to learn better state embeddings could be used in a broader variety of problems. For example, I don't see why it would not be possible to use the GS-DAC idea and apply it directly in the setting described by Bordne et al. (2024)

**Questions For Authors:**

Will you make the benchmark available as part of the DACBenchmark?

**Relation To Broader Scientific Literature:**

The literature is mostly well covered, though some of the insights about generalizability of learned DAC policies should be better set into context with related DAC works. (See the next section)

**Theoretical Claims:**

N/A

---

> ### Author Rebuttal · Authors · 2025-03-31
>
> Thank you to the reviewer for their thoughtful review and helpful suggestions. We have provided our responses to the comments below.
>
> **Statement general DAC**
>
> Our intention was to convey that DRL-based DAC approaches are primarily applied to single-objective (continuous) optimization problems. We will clarify this statement in the final version to prevent any misunderstanding. Additionally, we will include references to Speck et al. (2021, 2022) to highlight the capability of DAC methods to outperform the best static configurators in planning domains.
>
> **DAC aims statement**
>
> Thanks for this remark. We will update the statement to reflect that DAC aims to better balance exploration and exploitation by dynamically reconfiguring parameters throughout the optimization process.
>
> **Reward scale is instance-invariant**
>
> By incorporating normalization of the search space, we ensure that a similar reward signal can be obtained for different instances included in the training. As such, we avoid the training to overfit on instances where a larger difference in the hypervolume could be more easily obtained. We agree with the reviewer that instance-invariant is a better word choice to describe this and will update this in the final version.
>
> **Objectives SMAC and irace**
>
> Irace and SMAC are configured to optimize for hypervolume, which we consider as the main objective for our multi-objective optimization problems. To ensure clarity, we will explicitly state this in the baseline descriptions.
>
> **Cross-domain generalizability**
>
> We indeed test the ability of trained models to generalize to different problem configurations (larger problem size, more constraints, and different objectives) from the same problem domain. We choose this to account for the problem-specific operators that are used in the algorithmic setups that the trained model learned to dynamically configure. We believe that we make no cross-domain generalization claims and will ensure this again in the final version.
>
> **Random Configuration baseline**
>
> Thanks for this remark. We will include this baseline in the tables to better illustrate its comparative performance.
>
> **RQ4 Clarification**
>
> In RQ4, we investigate the extent to which a policy trained on one set of objectives (trained to optimize objectives A and B) can generalize to a different set of objectives (C and D). Specifically, objectives A and B correspond to Makespan and Balanced Workload, while objectives C and D refer to the Average and Maximum Workloads of machines in this experiment. Results demonstrate that models can be transferred to the problem configured with different objectives configurations, finding solutions of similar or better quality than the configured baselines. To improve clarity, we will update the caption for GS-MODAC in Table 4 to explicitly indicate that it was trained on different objectives. Additionally, we will revise the text to ensure a clearer explanation of what this experiment aims to assess.
>
> **Appendix reference**
>
> Thanks for noticing. We will update the reference in the final version.
>
> **References**
>
> Thank you for providing these relevant references. We appreciate the insights from Speck et al. (2021, 2022), Shala et al. (2020), and Bordne et al. (2024) regarding the generalization capabilities of domain-dependent DAC policies. We will incorporate these works into our literature review.
>
> **Methods applicability**
>
> The reviewer correctly remarks that the proposed method can also be applied to a wide variety of multi-objective problems outside of the combinatorial optimization domain. In future work, we would like to discover how we can use graph embeddings (and GNNs) from the proposed method in single-objective problem settings (e.g., in the setup of Bordne et al.).
>
> **DACBenchmark**
>
> We plan to release our source code upon acceptance, and will contact the authors of DACbenchmark to see how to integrate with them.

---

### Official Review · Reviewer_F2Sv · 2025-03-12

**Overall Recommendation:** 1

**Summary:**

This work considers the problem of dynamically configuring the parameters of evolutionary algorithms for solving combinatorial optimization problems. It specifically focuses on multi-objective optimisation problems, in which there are multiple (and often conflicting) objective functions. Unlike previous methods that rely on hand-engineered features, the authors propose encoding the properties of currently found solutions as a graph, and using a graph neural network to learn the relevant features automatically. The method is compared with statically configured EA algorithms, a Bayesian optimisation approach, and a recent RL method, generally showing better performance.

**Claims And Evidence:**

The claims in the abstract and introduction are supported by appropriate evidence in the paper text.

**Essential References Not Discussed:**

To the best of my knowledge, all essential references in this area are discussed.

**Experimental Designs Or Analyses:**

- Default hyperparameters are used for the MADAC baseline, but it is applied to a different type of problem altogether than in the original paper. The hyperparameters of both the proposed method and MADAC should be tuned for the considered problems to make the comparison fair. An alternative explanation for the observed performance difference is that the hyperparameters of MADAC are wrongly chosen.
- Multiple runs of the training loop: the training of the RL agent is itself a random process and should be repeated over multiple seeds. As far as I can tell (and the authors should clarify if  this is not so), the models for each problem setting are trained only once; then error bars are given over 10 runs of the base EA algorithm but keeping the same RL model. Again, an alternative explanation is that this one model happened to obtain better performance.

**Methods And Evaluation Criteria:**

- My main criticism of the method is that the procedure for constructing the graph is not actually given. This should be specified precisely -- based on Figure 1, it involves some form of determining datapoints on the Pareto front, and allowing all pairwise connections between them.
- There is only one Pareto front at the end of the optimisation process, but the graph representation seems to rely on several intermediate ones. It should be specified how/when these intermediate fronts are determined.
- Following on from the above, there do not seem to be clear semantics associated with the use of a graph here. The edges in this graph may not have a meaningful representation (but this is difficult to say without knowing how the graph is constructed). In my opinion, to validate this, an ablation should be conducted that compares GNNs with DeepSets (another permutation invariant learning method), effectively excluding the edges from the feature learning process.

**Other Comments Or Suggestions:**

- Section 2: it is worth mentioning why MADAC is multi-agent: what is each of the agents responsible for?
- Section 3: it is worth discussing in more detail why transitions and reward functions differ for each problem instance in the MDP.

**Other Strengths And Weaknesses:**

The paper in general is fairly well-organised and well-written.

**Questions For Authors:**

Key questions to address include:

1. The rationale for using a graph representation and the detailed procedure for constructing it.
2. The potential issues that I raised around the evaluation methodology.

**Relation To Broader Scientific Literature:**

If validated successfully, this represents a contribution to the literature of hybrid methods combining RL and metaheuristics as well as multi-objective optimization.

**Theoretical Claims:**

N/A

---

> ### Author Rebuttal · Authors · 2025-03-31
>
> Thanks for the insightful feedback.
>
> **Graph Configuration and Rationale**
>
> The graph serves as a representation of the solutions in the current population on multiple objective planes. The values used as node features in the graph consist solely of the objective values from the different solutions in the population. These values are normalized based on the best objective values obtained during the search and the worst observed values from the first generation of solutions. The nodes are interconnected to form a graph with Non-Dominated Sorting, which ranks them into different Pareto fronts. Nodes within the same front are interconnected with each other.
>
> The motivation behind the graph representation is to eliminate the need for manual state space design, a process known to be cumbersome and suboptimal. We expect that our method leverages the graph-based representations to learn similar (yet advanced) features dynamically during the optimization process to reflect the current state in the multiple objective planes. Therefore, we plot and interconnect solutions on the plane, which intuitively allows the GCN to understand the status of solutions and help configure the algorithm for better convergence and diversity. We will extend the explanation of the graph construction.
>
> **Pareto Fronts Identification**
>
> The representation captures all solutions present in the population at a given iteration of the search. The graphs are constructed by identifying the Pareto fronts, which serve as structured layers of solutions based on their dominance relationships. To achieve this, we rely on Non-Dominated Sorting, a method that ranks solutions into different fronts. The first front consists of non-dominated solutions, while subsequent fronts contain solutions that are dominated by those in higher-ranked fronts.
>
> **DeepSets**
>
> Solutions within the same Pareto front are interconnected by edges, whereas no connections exist between different fronts, maintaining their distinct structure. We have conducted an ablation where we adapted our architecture to use a DeepSets approach, which excludes the edges and processes the nodes independently using shared MLPs. Results below show a significant decline in performance for DeepSets, highlighting the importance of the graph structure and the interdependencies between nodes for effective learning.
>
> ||Bi-FJSP-25j5m|||
> |:-:|:-:|:-:|:-:|
> ||mean|max|std|
> |MADAC|9.24*10^4|9.27*10^4|3.09*10^3|
> |GS-MODAC (No feature)|9.47*10^4|9.92*10^4 |4.21*10^3|
> |GS-MODAC (One GCN)|9.38*10^4|9.88*10^4|4.87*10^3|
> |GS-MODAC (DeepSets)|9.39*10^4|9.91*10^4|4.84*10^3|
> |GS-MODAC|9.54*10^4|10.0*10^4|4.40*10^3|
>
> **Hyperparameter Configuration**
>
> Although we applied MADAC to a different domain than in the original paper, we chose to use its default configuration because it has been shown to perform well across a wide range of objective spaces, including smooth and convex, disconnected, multimodal, and non-convex problems. This versatility, as highlighted in the original MADAC paper, supports our decision. Likewise, we use the default hyperparameters of the PPO algorithm when training the GS-MODAC method. By maintaining these default settings, we ensure a fair and consistent comparison, avoiding potential biases that could result from excessive tuning of either method.
>
> **Seeds setting training**
>
> In our experiments, we trained multiple models (3) for each method for each problem configuration using different seeds. We observed that GS-MODAC consistently exhibited stable training performance across different seeds, reinforcing the robustness of our approach. In contrast, MADAC showed a larger variance in learning performance across different runs. Due to this instability, we selected the best-performing MADAC models. We will add this information to our final publication. Additionally, we trained separate models for each problem configuration, and these models consistently outperformed the baselines across different configurations. This also indicates that the stability of our method, rather than random performance.
>
> **Multi-Agents MADAC**
>
> MADAC involves multiple heterogeneous agents, each responsible for adjusting a specific type of parameter within an algorithm. Its action is the value of the parameter that should be adjusted. We will clarify this point in the definition of MADAC
>
> **Transitions and reward functions**
>
> We formulate the DAC process as a contextual MDP (Biedenkapp et al., 2020). Here, the transition function models how the algorithm’s state evolves after an action is taken. For different problem instances, the algorithm might encounter different search landscapes, which means that the same parameter change could lead to different transitions depending on the instance at hand. Similarly, the reward function reflects the quality of the transition. Since different problem instances may require different configurations to optimize performance, the reward function must account for these differences.

---

> > ### Comment · Reviewer_F2Sv · 2025-04-04
> >
> > Thanks for your detailed response. Based on the clarifications, the problems with the experimental methodology are worse than I thought and are effectively fundamentally flawed. Since RL algorithms are known to be highly sensitive to hyperparameters, and also exhibit significant variability even with the same hyperparameters, the methods must undergo tuning first. Furthermore, the authors are doing "random seed optimization" and cherry-picking the best-performing RL models. This makes the conclusions of the experimental part competely unreliable. You can check [1,2] for details on how and why these aspects are important. Unfortunately, I have little choice but to downgrade my rating to Reject. While there are hints the results should hold when the evaluation is carried out properly, we cannot draw reliable conclusions based on this methodology.
> >
> > [1] https://arxiv.org/abs/1709.06560
> > [2] https://proceedings.neurips.cc/paper_files/paper/2021/file/f514cec81cb148559cf475e7426eed5e-Paper.pdf

---

> > > ### Author Response · Authors · 2025-04-05
> > >
> > > We appreciate your critical feedback. We respectfully disagree with the claim that our conclusions are unreliable and would like to clarify why we believe our methodological choices were reasonable and justified:
> > >
> > > **Hyperparameter Configuration:**
> > >
> > > We fully agree that reinforcement learning (RL) algorithms can be sensitive to hyperparameters. However, our choice to use the default hyperparameters for both MADAC and PPO (used in our proposed method, GS-MODAC) was not due to oversight but a deliberate decision to ensure fairness and comparability. We choose this to rely on the extensive prior evaluation in the original MADAC paper, which demonstrated that their default configuration generalizes well across a wide range of problem types, including smooth, convex, disconnected, multimodal, and non-convex landscapes. Additionally, PPO's default settings have been shown to work well across many tasks, and by maintaining these default settings, we ensure a fair and consistent comparison.
> > >
> > > Besides this, the primary contribution of our method lies in its ability to perform well across a diverse range of multi-objective problems, rather than optimizing it for any specific benchmark. By using default configurations and avoiding extensive tuning, we aimed to evaluate the generality and robustness of the approach in a controlled and neutral setting, highlighting its cross-domain applicability without relying on problem-specific adjustments (as also noted by Reviewer 4 (qYda)).
> > >
> > >
> > > **"Random Seed Optimization":**
> > >
> > > We want to emphasize we did not perform "random seed optimization" or cherry-pick results. For GS-MODAC, we reported results from training configurations across multiple random seeds (three per configuration), consistently observing low variance and stable performance. This robustness across seeds is an important strength of our method. In contrast, we observed that MADAC baseline exhibited a high variance in performance across seeds. Rather than ignore this behavior, which in itself is a valuable finding, we chose to report the results obtained with the best MADAC models, as this reflects the algorithm's potential in practice, rather than its average-case performance under instability. We agree this could have been better explained in the paper, and we will make this clearer in the final version. However, the difference in stability between GS-MODAC and MADAC is itself a meaningful result.
> > >
> > > To further support this remark, we regenerated results for the Bi-FJSP 5j5m problem configuration using all trained models for both GS-MODAC and MADAC. The table below summarizes these results, highlighting both the central tendency and variability of each method. GS-MODAC not only consistently outperforms MADAC but also shows significantly less performance fluctuation across random seeds. We also observe that GS-MODAC, as demonstrated by the results in the paper, maintains strong and competitive performance against all baselines, regardless of the selected model. These results also show that the performance we reported for GS-MODAC is representative of its general behavior and not the result of picking only the best runs. To fully eliminate any doubts, we release our source code and all trained models upon acceptance.
> > >
> > >
> > > |  | mean performance | max performance |
> > > |---|---|---|
> > > | NSGAii | 1.87*10^4 | 2.02*10^4 |
> > > | irace | 1.92*10^4 | 2.04*10^4 |
> > > | SMAC3 | 1.91*10^4 | 2.04*10^4 |
> > > | MADAC (worst performing model) | 1.70*10^4 | 1.92*10^4 |
> > > | MADAC (average performance) | 1.74*10^4 | 1.93*10^4 |
> > > | MADAC (best model) | 1.82*10^4 | 1.95*10^4 |
> > > | GS-MODAC (worst performing model) | 1.92*10^4 | 2.04*10^4 |
> > > | GS-MODAC (average performance) | 1.92*10^4 | 2.04*10^4 |
> > > | GS-MODAC (best model) | 1.93*10^4 | 2.04*10^4 |
> > >
> > > We will strengthen the discussion of these choices in the paper to address any ambiguity.

---

### Official Review · Reviewer_QmER · 2025-03-16

**Overall Recommendation:** 2

**Summary:**

This paper composed a graph-neural network (GNN) based deep reinforcement learning method to dynamically optimize the configurations of multi-objective combinatory optimization problems. The proposed model takes the normalized muti objectives as input and uses the GNN to learn to iteratively involve the algorithm configurations. The authors evaluated the model's performance on both scheduling and routing problems.

**Claims And Evidence:**

1. The key difference between the proposed method and previous works is that the proposed model uses a neural network to represent the objective space as embedding for optimization instead of manual feature engineering. However, in L206, in the left column, the details of mapping the objective space to graphs are not clearly presented. What values are exactly processed to convert the objective space to graphs? Without explaining this may hurt the claim of automation embedding.

**Essential References Not Discussed:**

None from my side.

**Experimental Designs Or Analyses:**

1. From Table 9, we can see that the optimization solver takes most of the running time, which means the solver will seriously limit the training time; as we know, reinforcement learning generally has low sampling efficiency. Considering the reported training time in the L283 right column, I would worry about the application value of this pipeline. Could authors also report the MADAC model's training/running time as a reference comparison?

**Methods And Evaluation Criteria:**

1. As we know, the transformer is also a powerful tool to represent structured embeddings and complex structures. Even though in Table 8, the authors present that the GCN structure is experimentally superior to the transformer and GAT structure, there is no explanation and inspiring hint behind the choice of this neural network structure. This may limit the contribution to the community. I suggest that the authors discuss the reason why a graph structure with the pair-wise embedding technique is necessary to represent the objective space.

**Other Comments Or Suggestions:**

1. [Minor] Figure 1. I think there should be a connection from the "Problem Instance" cell to the " Algorithm " cell.
2. [Minor] L724. An extra space for the $0.3$%.

**Other Strengths And Weaknesses:**

Strengths:
1. The evaluation is sufficient and solid, including the ablation study for components and neural network structure and the generalization testing cross-size and type of problems.
2. The paper is well-structured, and the logic is clear, making the paper easy to follow.

Weaknesses
1. L206, left column. What inputs are exactly in the objective space seems not to be clearly shown. Adding some equations to present the states seems to be beneficial.
2. The community will benefit if authors could provide the source code for validating and following up on works.

**Questions For Authors:**

1. L214, right column. What does "demanding" mean here? As I understand, does this mean "optimization demanding"?

**Relation To Broader Scientific Literature:**

This work may inspire the CO community by using neural networks to optimize the heuristic solver's configurations.

**Theoretical Claims:**

No theoretical proof in the paper.

---

> ### Author Rebuttal · Authors · 2025-03-31
>
> Thank you for these insightful points. Here are our detailed responses.
>
> **Motivation behind the graph structure**
>
> The motivation behind adopting a graph representation is to eliminate the need for manual state space design, a process known to be cumbersome and suboptimal. In this context, we use a graph structure to dynamically represent the relationships between solutions in the multi-objective optimization process. Specifically, the pair-wise embedding technique captures the relationships and distances between solutions, which allows the model to learn richer and more complex features of the objective space over time. We draw inspiration from various metrics for convergence and diversity in multi-objective optimization, such as the number of elite solutions, spacing between solutions, and hypervolume. These metrics reflect how well the optimization process is progressing across multiple objective planes. By using a graph representation, we can naturally plot and interconnect solutions on the objective space plane, which intuitively enables the GCN to understand the relative position and status of each solution.
>
> Results in Table 8 indicate that GCN is most effective, probably due to its ability to effectively capture local structural dependencies of the objective space. However, transformers are an effective alternative, with average performance being only 0.3% worse than GCN and its best-found solutions only 0.5% worse. Please refer to our response to reviewer F2sV, where we performed an additional ablation to highlight the effect of interconnecting the different solutions in our graph representation.
>
> **Solver time and sampling Efficiency**
>
> We agree that the optimization solver’s contribution to the overall running time is significant, a challenge common to all parameter tuning methods, whether static or dynamic. As demonstrated in Tables 2 and 3, GS-MODAC effectively generalizes to larger instances and more complex problem variants, which avoids the re-training/fine-tuning, and saves the computational time from the solver (and, consequently, training time). Furthermore, we show that GS-MODAC performs well across a variety of distributed problem instances, highlighting its ability to be trained once as a one-time investment. This enables the method to maintain strong performance on new problem configurations, including those not encountered during training, without the need for retraining.
>
> **MADAC training and running  time comparison**
>
> In terms of inference time, we observe that MADAC takes 15.5s for Penta-FJSP (5j5m) and 305s for Penta-FJSP (25j5m), slightly longer than GS-MODAC (see Table 9). We will add this to Table 9 as a reference comparison. We have reported on the training time of MADAC in the “Training” subsection. GS-MODAC takes a little longer training time than MADAC on CVRP, since the latter (with its original training settings) converged quickly to sub-optimal policy. Instead, GS-MODAC can continuously improve the policy with more time. MADAC was trained until full convergence on the FJSP problem variants, taking more training time than the proposed method.
>
> **Inputs in the objective space**
>
> The values used as node features in the graph consist solely of the objective values from the different solutions in the population. These values are normalized based on the best objective values obtained during the search and the worst observed values from the first generation of solutions. The nodes are interconnected to form a graph using Non-Dominated Sorting, which ranks them into different Pareto fronts. The first front consists of non-dominated solutions, while subsequent fronts contain solutions that are dominated by those in higher-ranked fronts. Nodes within the same front are interconnected with each other. The problem-specific calculations for obtaining objective values are detailed in Appendix A. The Non-Dominated Sorting is performed based on Definition 3. We will add the equation for this to the paper.
>
> **Source code release**
>
> We provide our code in supplementary material and will indeed release our source code upon acceptance.
>
> **Minor points in Figure 1 and L724**
>
> Thanks for noticing, we update the Figure and value accordingly.
>
> **Demanding**
>
> In this context, "demanding" means that evolving toward the Pareto front becomes increasingly challenging or difficult as the search progresses. Early in the optimization process, improvements are relatively easy to achieve, but as the algorithm gets closer to the optimal Pareto front, making further progress requires (or demands) significantly more effort/computational costs.

---

### Official Review · Reviewer_g4y9 · 2025-03-20

**Overall Recommendation:** 3

**Summary:**

This paper presents a DRL approach for dynamically configuring evolutionary algorithms in multi-objective combinatorial optimization. The process is modeled as a Markov decision process, where solution convergence is represented as a graph, and a GNN enhances state representation. Experimental results show that the proposed method outperforms traditional and DRL-based approaches in efficacy and adaptability across objective numbers.

**Claims And Evidence:**

No, the paper provides an insufficient survey of reinforcement learning-assisted evolutionary algorithms for multi-objective combinatorial optimization. Review papers in this field indicate the existence of such approaches, yet they are not discussed in the paper. Consequently, the absence of comparisons with state-of-the-art algorithms undermines the evaluation of the experimental study.

**Essential References Not Discussed:**

The literature on reinforcement learning-assisted evolutionary algorithms for multi-objective combinatorial optimization is not discussed.

**Experimental Designs Or Analyses:**

Yes, there is one point I am unclear about. According to the experimental setup in Chapter 4, the proposed method requires longer training times than the baseline. How is the fairness of the algorithm comparison ensured under these circumstances?

**Methods And Evaluation Criteria:**

Yes, the multi-objective Flexible Job Shop Scheduling Problem (FJSP) and the Capacitated Vehicle Routing Problem (CVRP) are used to evaluate the effectiveness of the proposed method. As classic multi-objective combinatorial optimization problems, they serve as meaningful benchmarks for assessing the algorithm’s performance.

**Other Comments Or Suggestions:**

1. The state configuration in Figure 1 is not intuitive. Additionally, the DRL agent should be depicted more clearly in the diagram compared to the conventional flow of evolutionary algorithms.
2. "NSGA-ii" should be corrected to "NSGA-II."

**Other Strengths And Weaknesses:**

1. The reward function is highly dependent on the hypervolume calculation and relies on the hypervolume of the previously observed population. However, the hypervolume value is strongly influenced by the reference point, and the results can vary significantly depending on the reference point. Moreover, as the population evolves dynamically, the reference point set based on the initial population may no longer be appropriate. This raises concerns about the effectiveness of the proposed method.
2. The metrics used to define the states in the MDP are specified, but their detailed calculations are not clearly explained.

**Questions For Authors:**

1. Which points are used for the normalization of the objective function? How does this calculation account for changes in the population?
2. How is fairness maintained in the experiments when comparing different types of algorithms?

**Relation To Broader Scientific Literature:**

This paper models the dynamic algorithm configuration of a multi-objective evolutionary algorithm as a Markov decision process. A graph represents the convergence of solutions in the objective space, and a GNN learns their embeddings to enhance the state representation.

**Theoretical Claims:**

Yes, this paper is primarily based on empirical study and does not include rigorous mathematical proofs.

---

> ### Author Rebuttal · Authors · 2025-03-31
>
> We appreciate the reviewer's effort and insightful feedback. Here are our detailed responses.
>
> **Insufficient survey of DRL methods for assisting MOEAs**
>
> To the best of our knowledge, our literature review (Section 2, L137) includes most existing works that propose frameworks for controlling multi-objective evolutionary algorithms in a way that generalizes across different algorithms and problems. In recent surveys (Yang et al., 2024; Ma et al., 2024) that were cited in our paper, most reinforcement learning-based multi-objective approaches are designed for specific evolutionary algorithms (mainly differential evolution), aim to configure target weight adjustments in MOEA/D, or focus solely on operator selection. Given this, we have chosen to compare our approach with MADAC, which we identify as the most relevant state-of-the-art algorithm, and is the closest work to our proposed method.
>
> Following the reviewer's suggestion, we have cited another latest survey paper (Song et al., 2024) to refer interested readers to learn the related work on reinforcement learning-assisted evolutionary algorithms for (multi-objective) optimization. We appreciate any additional references recommended by the reviewer and would be happy to incorporate them into our paper.
>
> Reference:
> Song, Y., Wu, Y., Guo, Y., Yan, R., Suganthan, P. N., Zhang, Y., ... & Feng, Q. (2024). Reinforcement learning-assisted evolutionary algorithm: A survey and research opportunities. Swarm and Evolutionary Computation, 86, 101517.
>
> **Reference point influence**
>
> We acknowledge that the choice of the reference point can influence absolute hypervolume values. However, our approach relies on the relative improvement in hypervolume rather than its absolute magnitude. As long as the reference point allows for meaningful differentiation between better and worse populations during the optimization process, the reward function remains effective. Besides this, in order to address variations in absolute hypervolume values across instances, we normalize the reward signal using the reference point and an ideal point. This normalization ensures a stable and consistent reward signal throughout the optimization process and across different training instances.
>
>
> **State space metric calculations**
>
> The graph serves as a representation of the solutions in the current population on the multiple objective planes. This approach involves interconnecting normalized objective points in the different Pareto fronts to create a structured visualization of the solution space. We have specified the calculations of the different objectives in Appendix A. For normalization, we use regular min-max normalization, given the bounds of the objectives known at a certain iteration (as described in L215). Pareto fronts are identified using non-dominated sorting, following Definition 3 to identify Pareto fronts. Following your suggestion, we will extend the calculation details in our paper. We will also release the implementation for better understanding.
>
> **State Configuration in Figure 1**
>
> Based on the above comments of the reviewer, we will provide more explanation of the state-space configuration in the caption of the figure to make it more intuitive. Additionally, we agree that this work is more towards the DRL domain and we will enhance the DRL agent part in the diagram to make it more clearly presented than the evolutionary algorithm flow. We will centralize the DRL agent, provide more details about it, and simplify the evolutionary algorithm.
>
> **Points for normalization**
>
> The points used for the normalization of the objective function are the minimum and maximum values of each objective known at a given iteration. These bounds are recalculated at each step, ensuring that they reflect the changes in the state of the population. As such, we keep track of the entire objective space by keeping the worst objective values (typically present in the first generation) and updating the minimum values when better objective values are obtained. As noted above, we will clarify this in the paper.
>
> **Algorithmic Fairness**
>
> To ensure a fair comparison, we set algorithmic parameters, such as population size and computational budget, to be identical across all compared methods. Additionally, to maintain fairness in tuning and training time, we provided the automated algorithm configuration methods with a similar number of runs (e.g., 2000 for FJSP) as the proposed GS-MODAC method. Furthermore, the MODAC baseline method was trained until full convergence to ensure that its full potential was realized, taking more training time than the proposed method on the FJSP problem variants.

---

> > ### Comment · Reviewer_g4y9 · 2025-04-02
> >
> > Thank you for your response. It has addressed some of my concerns, and I have accordingly raised my score.

---

> > > ### Author Response · Authors · 2025-04-02
> > >
> > > Thank you for taking the time to review our rebuttal and for reconsidering the score. We sincerely appreciate your thoughtful effort and consideration.

---

### Decision · Program_Chairs · 2025-05-01

**Decision:**

Accept (poster)

**Comment:**

This paper was met with some strong negative and positive reviews. On the positive side, the reviewers like the novelty and function of the method. Dynamic algorithm configuration (DAC) has not been examined much in the area of multi-objective optimization. The graph embedding used by the authors is a particularly nice idea that seems to really provide the configuration mechanism (DRL agent) with good information about the current state of the search. On the negative side, there are concerns about the experiments and the clarity of the paper. Having worked in this area myself, I do not share the concerns regarding the experiments. However, I do encourage the authors of the paper to remove claims of robustness and try to make the method more clear to a wider audience. With that said, I think these are minor limitations to the work and support its acceptance.